# Investigating the effects of intersection flow localization in equivalent continuum-based upscaling of flow in discrete fracture networks

Maximilian O. Kottwitz[1,3], Anton A. Popov[1,3], Steffen Abe[2], and Boris J. P. Kaus[1,3]

[1]Johannes Gutenberg University, Institute of Geosciences, Johann-Joachim-Becher-Weg 21, 55128 Mainz, Germany
[2]Igem, Institut for Geothermal Ressourcemanagement, Berlinstr. 107a, 55411 Bingen, Germany
[3]Johannes Gutenberg University, M3ODEL - Mainz Institute of Multiscale Modeling, Staudingerweg 7, 55128 Mainz, Germany

**Correspondence:** Maxmilian O. Kottwitz (mkottwi@uni-mainz.de)

**Abstract.** Predicting effective permeabilities of fractured rock masses is a crucial component of reservoir modeling. Its often realized with the discrete fracture network (DFN) method, where single-phase incompressible fluid flow is modeled in discrete representations of individual fractures in a network. Depending on the overall number of fractures, this can result in high computational costs. Equivalent continuum models (ECM) provide an alternative approach by subdividing the fracture network into

a grid of continuous medium cells, over which hydraulic properties are averaged for fluid flow simulations. While continuum methods have the advantage of lower computational costs and the possibility of including matrix properties, choosing the right cell size to discretize the fracture network into an ECM is crucial to provide accurate flow results and conserve anisotropic flow properties. Whereas several techniques exist to map a fracture network onto a grid of continuum cells, the complexity related to flow in fracture intersections is often ignored. Here, numerical simulations of Stokes-flow in simple fracture intersections are

utilized to analyze their effect on permeability. It is demonstrated that intersection lineaments oriented parallel to the principal direction of flow increase permeability in a process we term intersection flow localization (IFL). We propose a new method to generate ECMs that includes this effect with a directional pipe flow parametrization: the fracture-and-pipe model. Our approach is compared against an ECM method that doesn't take IFL into account by performing ECM-based upscaling with a massively parallelized Darcy-flow solver capable of representing permeability anisotropy for individual grid cells. While IFL

results in an increase of permeability at the local scale of the ECM cell (fracture scale), its effects on network scale flow are minor. We investigated the effects of IFL for test cases with orthogonal fracture formations for various scales, fracture lengths, hydraulic apertures and fracture densities. Only for global fracture porosities above $30\%$, IFL starts to increase the systems permeability. For lower fracture densities, the effects of IFL are smeared out in the upscaling process. However, we noticed a strong dependency of ECM-based upscaling on its grid resolution. Resolution tests suggests that, as long as the cell size is

smaller than the minimal fracture length and larger than the maximal hydraulic aperture of the considered fracture network, the resulting effective permeabilities and anisotropies are resolution-independent. Within that range, ECMs are applicable to upscale flow in fracture networks.

# 1   Introduction

Discontinuities in rocks provide major pathways for subsurface fluid migration. Thus, fractured reservoirs are frequent targets
for oil, gas, or water production, geothermal energy recovery, and $CO_2$ sequestration. In addition, the safety of nuclear waste
disposals and subsurface contaminant transport crucially depends on the presence of fractures. Characterizing natural fracture
networks across scales and model fluid flow therein to predict their effective permeabilities has thus been a long-standing topic
of research (e.g., Long et al., 1982; Dershowitz and Einstein, 1988; Cacas et al., 1990; Neuman, 2005; de Dreuzy et al., 2012).
Numerical modeling of fluid flow is most accurately based on the Navier-Stokes equations (Bear, 1972). For a single phase
of incompressible and iso-viscous fluid in an iso-thermal system, they simplify to the Stokes equations if laminar flow con-
ditions are considered (i.e., Reynolds numbers below 1 -10). Assuming an impermeable rocks matrix, one can solve for the
velocity distribution resulting from prescribed pressure boundary conditions, allowing to determine the rocks effective perme-
ability utilizing Darcy's law for flow through porous media (e.g., Andrä et al., 2013b; Osorno et al., 2015; Eichheimer et al.,
2019, 2020; Kottwitz et al., 2020). Those so-called direct-flow modeling approaches crucially rely on a digital representation
of a rock that separates pore-space from the matrix, which results from high-resolution X-ray computed tomographies (Andrä
et al., 2013a; Cnudde and Boone, 2013). However, they are limited in maximum scannable size and respective trade-off to
numerical resolution, making them applicable to small scales only (nanometers to a couple of centimeters at most). At larger
scales (above a couple of centimeters), so-called continuum-flow approaches serve to model fluid flow, usually based on the
concepts for flow through porous media proposed by Darcy (Darcy, 1856). Instead of a representation of the medium's pore-
space, they require an initial hydraulic representation of the medium. This is given by prescribed effective permeabilities for
certain control volumes within the medium, which upscale hydraulic properties from smaller scales to observation scales. Thus,
the key of this so-called upscaling problem (e.g., Zhou et al., 2010; Hauge et al., 2012; Lie, 2019) is to adequately represent
the rock structure with an appropriate model of effective permeabilities, which for fractured rock masses is often cumbersome
due to their structural heterogeneity (Dershowitz and Einstein, 1988; Odling et al., 1999). The main problem is that acquiring
detailed natural fracture data in 3D is intricate, as seismic imaging techniques suffer from resolution limits (Cartwright and
Huuse, 2005; Malehmir et al., 2017), preventing a multi-scale structural assessment of individual features in fracture forma-
tions. Hence, outcrop (2D) and borehole (1D) studies are the only possibilities to acquire detailed natural fracture data, despite
their reduced dimensionality (Lei et al., 2017), and acquiring deterministic knowledge of all individual structures in a fracture
formation is impossible. Due to this, the discrete fracture network (DFN) method has been extensively used as a conceptual
framework to provide statistically-based approximations of real fracture networks for decades (Long et al., 1982; Cacas et al.,
1990; Bogdanov et al., 2003; Darcel et al., 2003; Xu and Dowd, 2010; Davy et al., 2013; Maillot et al., 2016). In this approach,
each fracture in a given network is represented with a reduced order object (lines in 2D and discs or rectangles in 3D) with
a prescribed location, size, and orientation. Naturally measured structural properties like size- and orientation-distributions
(Odling et al., 1999; Healy et al., 2017) as well as fracture density and spacing (Ortega et al., 2006) serve as quantitative ba-
sis to prescribe their geometrical properties (e.g., Hyman et al., 2015; Alghalandis, 2017). The hydraulic response to pressure
changes of each individual fracture is then parametrized with the cubic law (Snow, 1969; Witherspoon et al., 1980), relating the

fractures effective permeability to its aperture. In reality, surface roughness, fracture closure as well as fluid-rock interactions (e.g., erosion or crystal growth) cause deviations from the parallel-plate assumption (Brown, 1995; Oron and Berkowitz, 1998; Méheust and Schmittbuhl, 2000). Semi-empirical functions derived from numerical simulations in rough-walled fractures with quantified statistics of the aperture field (e.g., Patir and Cheng, 1978; Brown, 1987; Renshaw, 1995; Zimmerman and Bodvarsson, 1996; Mourzenko et al., 2018) serve as corrections to the cubic-law, if the fractures internal correlation length-scale is significantly smaller than the size of the considered fracture (e.g., Méheust and Schmittbuhl, 2003; Kottwitz et al., 2020).

A large number of numerical methods to compute effective permeabilities of fractured media have been developed (see reviews of Jing, 2003; Berre et al., 2019), all relying on (modified) cubic-law assumptions. Improved discretization techniques with individual fracture treatment (DFN method), inclusion of matrix properties in multi-dimensional meshes (discrete fracture and matrix - DFM - method) or multi-continuum methods and come at the cost of high computational expenses. Discretizing the fractured media as equivalent single continuum blocks significantly reduces the computational effort at comparable numerical accuracy (Hadgu et al., 2017).

According to Long et al. (1982) and Oda (1985), fractured rocks behave similar to porous media and can be represented by a positive definite permeability tensor (Chen et al., 1999) as long as the considered system behaves like a representative elementary volume (REV) (Bear, 1972), i.e., its effective properties (permeability or porosity for example) are more or less homogenous at the reference scale of the system. Due to the multi-scale character of fracture systems (e.g., Bonnet et al., 2001; Davy et al., 2006), determining the required homogenization scale is difficult, as distinct larger features may dominate overall flow. Thus, a discrete representation of all fractures in a network given by the DFN method is essential to adequately capture that multi-scale character. La Pointe et al. (1995), Jackson et al. (2000), Svensson (2001), Leung et al. (2012) and Hadgu et al. (2017), among others, have however showed, that representing a DFN with a grid of equivalent continuum blocks of sizes lower than the REV yields similar flow results, if resolved sufficiently, and thus reproduces the overall flow-behaviour of the DFN method. This highlights that continuum methods for flow modeling in fractured rocks are not restricted to REV scales and can thus be used equivalently to the DFN method.

Several techniques to generate equivalent continuum models (ECM) of DFNs have been developed in 2D (Reeves et al., 2008; Botros et al., 2008; Rutqvist et al., 2013; Chen et al., 2015) and 3D (Hadgu et al., 2017; Sweeney et al., 2020), whereby the so-called Oda method (see Oda, 1985) is used to formulate permeability tensors of grid cells that intersect fractures. There, the permeability tensor is aligned with the orientation of the intersecting fracture, and the permeabilities of the individual fractures are summed up if multiple fractures intersect one cell, yielding a positive definite, fully anisotropic tensor (e.g., Chen et al., 1999). The groundwater-flow equations for porous media (Bear, 1972), i.e., Darcy's law (Darcy, 1856), are then used to simulate laminar, steady-state, single-phase flow to compute effective permeabilities of the medium. There, current issues in commonly used 3D flow solvers, such as PFLOTRAN (Lichtner et al., 2015), are a lack of a fully anisotropic permeability representation at the local cell level. So-called stair-case patterns are the direct consequence of these simplifications, which introduce artificially prolonged flow-paths, especially in transport simulations, which have to be compensated for (e.g., Reeves et al., 2008; Botros et al., 2008; Sweeney et al., 2020) when predicting effective permeabilities of fractured media. On the other hand, MODFLOW (McDonald and Harbaugh, 1988) introduced support for local permeability anisotropy but not within

a massively parallelized framework, making it difficult to conduct large numbers of high-resolution simulations. However, assessing permeabilities in a Monte-Carlo-like framework (e.g., de Dreuzy et al., 2012) is necessary to explore the variance of hydraulic system properties induced by stochastically generated input-data. Hence, a flow-solver that combines the advantages of local permeability anisotropy and massive parallelization should be beneficial for numerical permeability assessments of fracture networks.

Next to these issues, this study focuses on an often ignored but potentially important aspect in fracture network modeling given by the complexity of fracture intersection flow. To our knowledge, only a few studies have presented 3D flow simulations within fracture intersections (Zou et al., 2017; Li et al., 2020), revealing the fact that flow velocities will increase within the fracture intersections compared to the fractures itself (shown by increasing Péclet numbers within the intersections). Theoretically, this effect should increase if the direction of the applied pressure gradient is aligned with the orientation of the intersection. As a consequence, the systems effective permeability should increase by a certain amount due to a local permeability increase within the intersection. To demonstrate that, we systematically conduct 3D numerical simulations of Stokes flow within differently oriented, planar fracture crossings to analyze the permeability increase caused by intersection flow localization (IFL). Using these results, we extend the current state-of-the-art methodology for equivalent continuum representations of DFNs to account for IFL in a quantitative manner and analyze its impact on effective permeability computations at fracture and network scales. There, it is still unclear at which level of detail the ECM has to be discretized to conserve the structural complexity of the DFN, as aforementioned stair-case patterns and artificial connectivity cause resolution dependencies. Subsequently, resolution tests are performed on two DFN test-cases with a newly developed, massively parallelized, and high-performance computing (HPC) optimized finite element Darcy-flow solver capable of handling fully anisotropic permeability tensor cells. By that, we consistently investigate the upscaling capabilities of the ECM method, which is frequently used for effective permeability predictions in fractured porous media.

## 2 Fracture intersection flow modelling

Fluid flow in porous and fractured media is described by the well-known Navier-Stokes equations (Bear, 1972). It is commonly assumed that sub-surface flow in fractures ranges in the laminar regime, i.e. Reynolds numbers below unity (Zimmerman and Bodvarsson, 1996). Assuming the flowing fluid to be incompressible, isoviscous and the impact of gravity to be negligible, steady-state flow at constant temperature is defined by Stokes momentum balance (eq. 1) and continuity (eq. 2) equations (Bear, 1972):

$$\mu \nabla^2 v = \nabla P, \tag{1}$$

$$\nabla \cdot v = 0, \tag{2}$$

with the fluid's dynamic viscosity $\mu$, pressure $P$ and velocity vector $v = (v_x, v_y, v_z)$. $\nabla$, $\nabla \cdot$, and $\nabla^2$ denote the gradient, divergence, and Laplace operator for 3D Cartesian coordinates, respectively.

Here, the 3D staggered grid, finite-difference code LaMEM (Kaus et al., 2016) is used to solve the coupled system of equations 1 and 2, utilizing PETSc (Balay et al., 2018) for HPC optimisation. Applying different absolute pressures on two opposing sides of a 3D voxel model representing the fractured or porous medium (e.g., a) or d) in figure 1) while setting the other boundaries to no-slip (velocity component normal to the boundary is zero) enables the prediction of the mediums directional permeability. After obtaining the steady-state solution, the volume integral of the pressure-gradient aligned velocity component $v_z$ (e.g., Osorno et al., 2015) is computed according to:

$$a)\bar{v} = \frac{1}{V}\int\limits_V |v_z|\,dz, \tag{3}$$

with domain volume V. Using Darcy's law for flow through porous media (Darcy, 1856), that relates the specific discharge $Q$ for a pressure drop $\Delta P$ along a distance $L$ according to:

$$Q = -\frac{kA\Delta P}{\mu L}, \tag{4}$$

with intrinsic permeability $k$ and cross-sectional area $A$ in combination with the fact that $Q = \bar{v}A$, the directional permeability $k_z$ is calculated by:

$$k_z = \frac{\mu \bar{v} L}{\Delta P}. \tag{5}$$

As demonstrated by Eichheimer et al. (2019); Kottwitz et al. (2020); Eichheimer et al. (2020), the numerical resolution has to be sufficiently high to produce a converged result. Generating every model at different levels of detail (e.g., $128^3$, $256^3$, $512^3$ and $1024^3$ voxels), ensures that the most accurate solution is obtained (as will be shown later by a comparison of errors to the result at largest resolution in plot b, figure 5). Figure 1 presents Stokes-flow in simple fracture intersections and highlights the IFL effect. If the fracture intersection is aligned with the principal flow direction (plot a) - c)), the velocity significantly increases within the intersection, resulting in higher directional permeabilities. In the opposite case, when the fracture intersection connects no-pressure boundaries (plot d) - f) ) and is thus oriented oblique to the flow direction, the flow velocity slightly disperses around the intersection, and the overall impact on the directional permeability is minor.

## 3  Permeability parametrization concepts

As the two main structural features (fractures and intersections) composing a fracture network differ significantly in terms of their hydraulics (figure 1), they require independent concepts to parametrize their permeabilities for formulating their effective grid block permeability tensor. For fractures, it is usual practice to use the cubic-law parametrization (e.g., Snow, 1969; Long et al., 1982), relating the specific discharge $Q$ through a void system between two parallel plates for a pressure drop $\Delta P$ along a distance $L$ according to:

$$Q = -\frac{wa_m^3\Delta P}{12\mu L}, \tag{6}$$

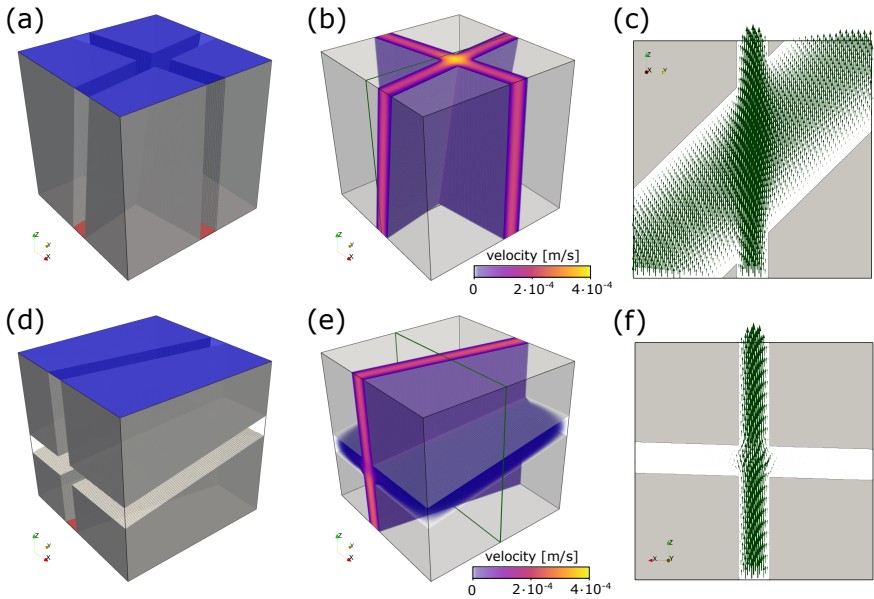

**Figure 1.** a) and d) show the binary voxel-models (impermeable matrix in transparent gray) for a fracture intersection that is orientated along and transverse to the flow direction, respectively. The red bottom face is the high-pressure boundary (0.02 $Pa$), the blue top face the low-pressure boundary (0.01 $Pa$), forcing the fluid to flow in $z$-direction. The orientations (arranged as dip-direction/dip) for the fracture pair in a) are $f_1 = 100/90$, $f_2 = 190/80$, and $f_1 = 170/90$, $f_2 = 260/10$ for the fractures in d). The length of both cubes is 1 $cm$ and all fracture apertures are constant (1.25 $mm$). b) and e) visualize flow velocity distribution in the void space. e) and f) highlight velocity vectors within the intersections at slices indicated with green rectangles in b) and e), respectively.

with the fractures width $w$ and distance between the two plates, i.e. mechanical aperture $a_m$. Comparing this analytical solution with Darcy's law (eq. 4, cross-sectional area $A = wa_m$) leaves the intrinsic permeability of a fracture $k_f$ defined by:

$$k_f = \frac{a_m^2}{12}. \tag{7}$$

Natural fractures deviate from the assumptions of parallel plates, which is why $a_m$ in eq. 7 is commonly replaced with a hydraulic aperture $a_h$ that corrects the parametrization for fracture closure and surface roughness (e.g., Patir and Cheng, 1978; Brown, 1987; Renshaw, 1995; Zimmerman and Bodvarsson, 1996; Kottwitz et al., 2020). Yet, there is no ready to use parametrization concept tailored for fracture intersections. The simulations shown on figure 1 suggest that the flow in the intersection is approximately pipe-like. Then, the specific discharge $Q$ through a tube of radius $r_t$ and length $L$ is related by 160   the Hagen-Poiseuille flow solution through a pipe (e.g., Batchelor, 1967) according to:

$$Q = -\frac{\pi r_t^4 \Delta P}{8L\mu}. \tag{8}$$

Again, combining this equation with Darcy's law (eq. 4, cross-sectional area $A = \pi r^2$) results in the following expression for the intrinsic permeability of a pipe $k_i$:

$$k_i = \frac{r^2}{8}. \tag{9}$$

The apparent pipe radius should then be modified based on the intersection shape to calculate an equivalent hydraulic radius $r_h$ to compensate for the structural difference. As a first-order approximation, we use half the size of the hypotenuse in a right-angled triangle whose legs are given by the two intersecting apertures (called half-hypotenuse assumption in the following, see figure 2 for details). This delivers sufficiently good results, as will be demonstrated later (figure 6).

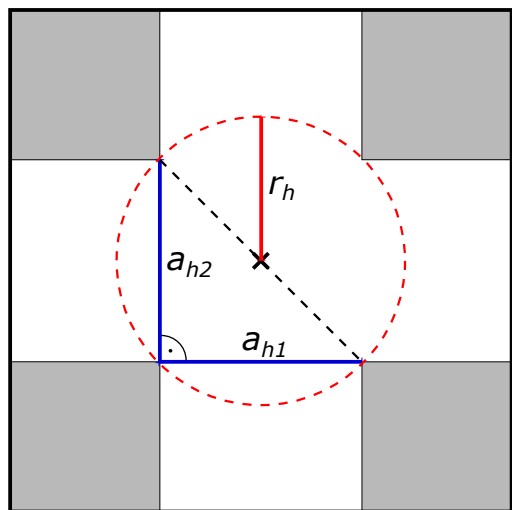

**Figure 2.** 2D Sketch of the half-hypotenuse assumption in an idealized rectangular fracture crossing (grey regions indicate rock matrix, white regions fracture pore space). The hydraulic apertures ($a_{h1}$ and $a_{h2}$) of both intersecting fractures are indicated with solid blue lines. The hypotenuse of the right-angled triangle with the two hydraulic apertures as legs is given by the black dashed line. The hydraulic radius $r_h$ (indicated by the solid red line) to approximate the radius of the pipe model is defined as half of the length of the hypotenuse.

## 4    Equivalent continuum representation of DFNs

The use of the ECM approach instead of the DFN method to predict the effective permeabilities of fractured media crucially depends on the capability to reflect the anisotropic flow properties at the scale of the continuum cells. Therefore, it is essential to integrate the geometry of a DFN into the generation procedure of the ECM instead of generating the grid cell conductivities in a stochastic manner (Hadgu et al., 2017). The accuracy of the ECM permeability prediction then depends on the resolution of the DFN-mapped continuum grid. Jackson et al. (2000) and Svensson (2001) already demonstrated that using cell sizes that

are larger than the average fracture spacing of the network introduces artificial connectivity and hence overestimates effective

permeabilities. Sufficient resolution of the continuum grid is therefore required to obtain comparable results with the DFN method (e.g., Botros et al., 2008; Leung et al., 2012).

To our knowledge, there is no approach to generate an ECM of a DFN that takes the effect of IFL (figure 1) into account. Thus, we will explain our new approach to generate continuum representations based on DFN structures - the fracture-and-

pipe model.

Generally, the DFN approach offers a straightforward way to characterize structurally complex fracture networks. Most commonly, every fracture is modeled as a geometric primitive (here a disc) with a prescribed length $l$, center coordinate $p_0$ and unit normal vector $\bar{n}$ defining its orientation. Based on this, fracture intersections can be calculated to define the backbone of the network. Here, fracture intersections are approximated with a line defined by two points $i_0$ and $i_1$, whereas the unit vector

$\bar{i}$ between the two points defines its orientation. The goal of the ECM method is to generate a 3D of computational cells that contains a symmetric, positive definite permeability tensor that is based on the fractures and their intersections. For simplicity, we prescribe a regular grid with constant $x$-$y$-$z$ spacing $\delta x$ instead of octree refined grids as utilized by Sweeney et al. (2020).

To map each individual fracture to its corresponding grid cells, we first assume a horizontal disc (normal vector $\bar{g} = [0,0,1]$) at center point $p_g = (0,0,0)$ with corresponding fracture radius $r$ ($r = l/2$) and represent it with an equally spaced set of points

in the $x$-$y$ plane $P_g$, with the condition $||P_g - p_g)|| \leq r$. By that, we obtain a constantly spaced grid of points representing the fracture in horizontal orientation, provided that the initial equal spacing of the points $\delta p$ is a small fraction of the cell size $\delta x$ to prevent gaps in the mapped 3D grid. Next, we seek the rotation matrix $R_f$ that aligns the current normal vector of the $x$-$y$ plane $\bar{g} = [0,0,1]$ with the actual normal vector of the fracture $\bar{n}$. Utilizing Rodrigues's rotation formula (Rodrigues, 1840) around the rotation axis $w = (\bar{g} \times \bar{n})/||(\bar{g} \times \bar{n})||$ (unit vector orthogonal to $\bar{g}$ and $\bar{n}$) yields the rotation matrix $R_f$ according to:

$$R_f = I + ||\bar{g} \times \bar{n}|| \, C + (1 - \bar{g} \cdot \bar{n}) \, C^2, \tag{10}$$

with $\times$, $\cdot$, and $||x||$ denoting the cross-product, dot-product and vector norm of $x$, respectively. $I$ represents the 3-by-3 identity matrix and $C$ the cross-product matrix of the rotation axis $w = [w_x, w_y, w_z]$:

$$C = \begin{bmatrix} 0 & w_z & w_y \\ w_z & 0 & -w_x \\ -w_y & w_x & 0 \end{bmatrix}. \tag{11}$$

Following this, $R_f$ is used to rotate the $3 \times n$ array of points representing the fracture plane $P_g$ ($n$ is the number of 3D points

in $P_g$) around $p_g$ and translate all points to the actual center point $p_0$ to produce a rotated set of points $P_r$ representing the fracture in its actual 3D position:

$$P_r = P_g * R_f + p_0, \tag{12}$$

where $*$ denotes matrix-matrix multiplication. By ensuring that the lower left corner coordinate of the rectangular grids bounding box is initially located at $(0,0,0)$ (this may require a translation of all center points to incorporate all fractures), we obtain

the grid-indices (i,j and k in $x$,$y$ and $z$-direction, respectively) of the fracture by dividing $P_r$ with the cell size $\delta x$ and rounding the results. Finally, we compute the individual anisotropic permeability tensor $K_{ijk}$ for the cells by using a parametrized

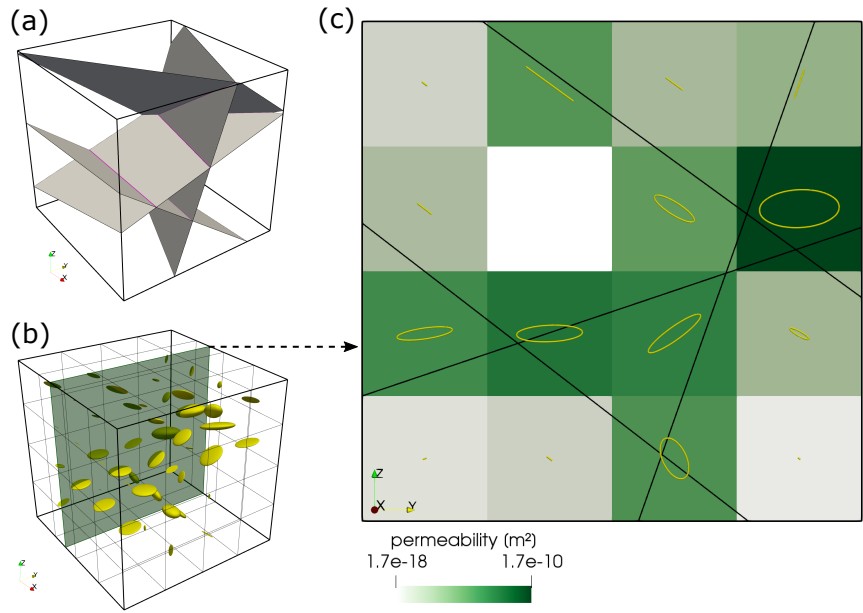

**Figure 3.** Workflow for generating an equivalent continuum model of a DFN. a) shows the input fracture network of 4 arbitrarily oriented fractures (gray) and their intersections (magenta). b) displays a grid of ellipsoids, each reflecting the shape of the permeability tensor in the equivalent continuum model of a) with a resolution of $4^3$ voxels. The size of the ellipses is scaled to the norm of the permeability tensor of the cell, such that larger ellipsoids denote higher permeabilities. The green plane in b) indicates the location of the 2D slice displayed in c). There, different green-intensities present the norm of the permeability tensor of each cell. Black lines denote fractures in 2D and yellow ellipses the x- and y-shape of the permeability tensor of each cell. Note, how the shape of the ellipse changes from being planar, if multiple fractures cross a cell.

fracture permeability value (eq. 7) and the rotation matrix $R_f$ according to:

$$K_{ijk} = \frac{V_f}{V_c} \, k_f \left( R_f \begin{bmatrix} 1 & 0 & 0 \\ 0 & 1 & 0 \\ 0 & 0 & 0 \end{bmatrix} R'_f \right). \tag{13}$$

$V_c$ denotes the cell volume ($\delta x^3$) and $V_f$ the fracture volume per cell, which is approximated by counting the number of
$P_r$ points per individual cell, multiplying it with the squared initial point spacing $\delta p$ and the hydraulic aperture $a_h$ of the fracture. Obviously, the accuracy of $V_f$ crucially depends on the initial point spacing of $P_g$ - the finer the spacing, the better the approximation of $V_f$. Plot c in figure 4 shows that the condition $\delta x / \delta p \geq 16$ delivers sufficiently constant permeability values. In case multiple fractures transect the same cell, the permeability tensors are summed, similar to Chen et al. (1999) or Hadgu et al. (2017). However, these cells need additional treatment as they incorporate fracture intersections. We follow the same
workflow as presented for individual fractures to map all previously found intersections to the grid cells and formulate their permeability tensors. A horizontal line of the same length as the intersection ($||i_1 - i_0||$), parallel to the $x$-axis is represented

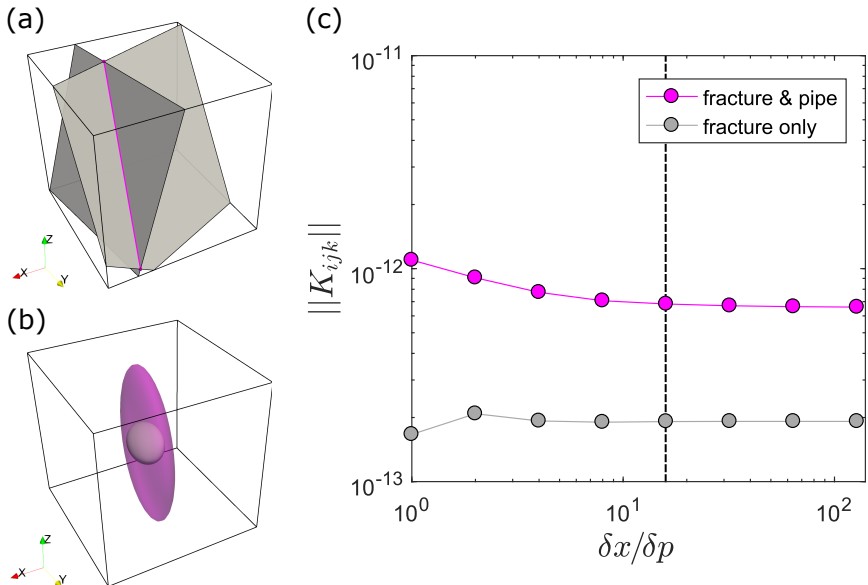

**Figure 4.** Fracture intersection caused changes of permeability tensor characteristics. a) shows a simple DFN structure of two arbitrary oriented fractures (grey) intersecting at a line (magenta). The cube length is set to $1\ m$ and the system origin is at $(0,0,0)$. The center point of the first fracture is located at $(0.4899|0.5685|0.5110)$ and its normal vector is given as $(-0.3195, 0.7894, 0.524)$. The second fractures center point is located at $(0.7604|0.5000|0.5000)$, whereas its normal vector is given by $(-0.9461, 0.1715, 0.2747)$. Both fractures have the same hydraulic aperture of $1 \cdot 10^{-3}\ m$ and both fully penetrate the system. The resulting intersection ranges from point $(0.6499|0.3086|1.000)$ to $(0.8003|1.000|0.0505)$ and its orientation is given by the unit vector $(0.1270, 0.5839, -0.8018)$. The hydraulic pipe radius resulting from the half-hypotenuse assumption is $7.0711 \cdot 10^{-4}$. b) visualizes the shape of the permeability tensor for an ECM model that considers only fracture permeability (grey, inside) and for the presented fracture-and-pipe model (transparent magenta, outside). The size of both ellipses is scaled with the norm of the resulting permeability tensor to provide comparability. c) presents the norm of the permeability tensor $K_{ijk}$ as a function of the ratio between the ECM grid spacing $\delta x$ and the initial point spacing $\delta p$ for the discretization approach described in section 4 (the fracture-and-pipe model) and an approach, where we didn't take the IFL parametrization into account (i.e., leaving out eq. 14 in the discretization procedure, hence the name fracture-only). The dashed black line denotes the condition $\delta p/\delta x \geq 16$, which is used to provide a correct approximation of the fracture and intersection volume per cell.

by a constantly spaced set of points (similar spacing as in the case of a fracture, i.e., $\delta p$). The mean point of the line is again located at $(0,0,0)$. We then calculate the rotation matrix $R_i$ (eq. 10) by using $\bar{g} = [1,0,0]$ and $\bar{n} = (i_1 - i_0)/||i_1 - i_0||$. After identifying the corresponding grid i,j, and k indices as described above, their permeability tensors are increased by using a

parametrized intersection permeability (eq. 9):

$$K_{ijk} = K_{ijk} + \frac{V_i}{V_c} k_i \left( R_i \begin{bmatrix} 1 & 0 & 0 \\ 0 & 0 & 0 \\ 0 & 0 & 0 \end{bmatrix} R'_i \right). \tag{14}$$

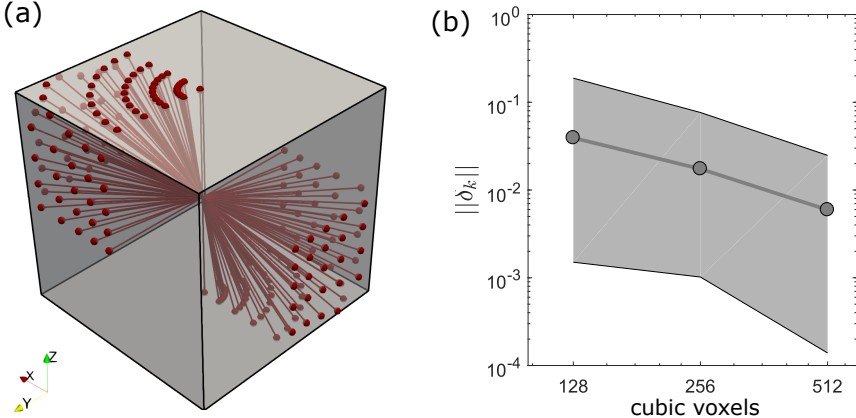

**Figure 5.** a) displays the location of all 100 intersection lineaments considered in the flow benchmark. 52 intersection configurations directly connect in- and outlets of flow (upper and lower $z$-face), whereas 48 connecting non-boundary flow faces. b) compares the numerically estimated permeability at highest resolution ($1024^3$ voxels) to the ones obtained at lower resolutions by calculating their error norms $||\delta_k||$ according to eq. 15. Gray dots represent the average error norm for all considered intersection configurations at resolutions lower than $1024^3$ voxels, and the light gray area highlights the range between minimum and maximum error.

$V_i$ represents the intersection volume per cell, which is again approximated by counting the number of $P_r$ points per cell and multiplying it with point spacing $\delta p$ and the term $\pi r_h^2$, whereas $r_h$ denotes the hydraulic radius of the pipe approximating the intersection. Figure 3 shows the resulting ECM structure with $4^3$ cells of an arbitrary complex DFN, generated with the presented approach. For certain fracture systems (ideally no more than two fractures that fully penetrate the system, e.g., plot a) in figure 4), the presented approach can be used to derive an analytical solution for permeability by setting $\delta x$ equal to the system size, resulting in a single permeability tensor for the whole system. Figure 4 demonstrates that incorporating the intersection as a pipe has a significant effect on the shape and absolute value of the permeability tensor at intersections, which could cause an overall permeability increase by almost one order of magnitude. However, the exact amount of permeability increase depends on the chosen hydraulic radius of the pipe, and the impact on the overall permeability at the network needs to be evaluated.

## 5   Fracture scale intersection flow benchmark

To test the half-hypotenuse assumption (see figure 2 for details) as a first-order approximation for the hydraulic radius of the pipe, we conduct a benchmark study in the following. The directional permeabilities of simple fracture crossings with varying orientations are calculated from high-resolutions Stokes-flow simulations (e.g., section 2) are compared to their analytically derived ECM single-cell counterparts (i.e., $\delta x$ is equal to the full system size $L$) using (1) the half-hypotenuse parametrization for intersection flow (fracture-and-pipe model) and (2) omitting this intersection flow parametrization (fracture-only model). For each intersection model, two fully persistent fractures with constant hydraulic apertures of $1.25\ mm$ are placed in a cube of

length 10 $mm$. Two fractures with a dip angle of 90° and dip directions separated by 90° (i.e., 90° and 180°) are consecutively rotated counter-clockwise by increments of 10° around the center of the cube until a total rotation of 90° is reached. This procedure is repeated nine times, whereas the dip angle of one of the two fractures is consecutively reduced by increments of 10° for each iteration. The dip angle of the remaining fracture is kept constant (i.e., 90) to maintain connectivity in the $z$-direction. This results in a total of 100 different intersection configurations (52 representing direct in- and outlets of flow, 48 connecting non-boundary flow faces), producing a wide variety of intersection orientations within two opposing octants in the cube (see figure 5 a for all generated intersection lineaments). For each configuration, we produce a binary voxel model (pore-space and matrix) of two crossing parallel plate fractures (similar to a) and d) in figure 1). Following the approach described in section 2, different pressures at the bottom and top boundary are applied to numerically estimate the directional permeability (setting the remaining boundaries to no-slip yields the vertical permeability component of the permeability tensor, $k_z$). We were systematically increasing the numerical resolutions of the Stokes-flow simulations ($128^3$, $256^3$, $512^3$ and $1024^3$ voxels) for each intersection configuration (resulting in a total of 400 HPC flow simulations) to determine whether the result at the highest level of detail represents a sufficiently converged solution. This is done by calculating the L2-error-norm $||\delta_k||$ according to:

$$||\delta_k|| = \left| \frac{k_x - k_{1024}}{k_{1024}} \right|, \tag{15}$$

whereas $k_{1024}$ represents the directional permeability obtained at the highest resolution (i.e. $1024^3$ voxels) and $k_x$ the directional permeability from simulations with lower resolution (i.e., $128^3$, $256^3$, $512^3$ voxels). The resulting average error norms for all 100 intersection configurations are plotted in figure 5 b, which demonstrate the convergence towards the numerical result at the highest resolution. With an average error norm of about 0.6 % and a maximum error of 2.4 % for simulations with $512^3$ voxels compared to the simulations at $1024^3$ voxels, we assume that the solution at $1024^3$ voxels represents a sufficiently accurate solution and can furthermore be used to benchmark the tensors generated with the ECM approach. Next, we follow the approach of section 4 to generate a single-cell permeability tensor of each intersection model, using a $\delta p/\delta x$ ratio of 16 and extract the vertical permeability component of the tensor ($k_{zz}$) and compare it with the one resulting from the Stokes-flow simulations. The results (figure 6) demonstrate that, if the intersection connects the two pressure boundary faces (intersection-to-flow-direction angle $\gamma \leq 40°$), the actual permeability obtained from the Stokes simulations is reasonably well reproduced with a small underestimation by the fracture-and-pipe model and heavily underestimated by the fracture-only approach (e.g., Hadgu et al., 2017). Using the half-hypotenuse assumption sufficiently integrates the effect of IFL at the scale of a continuum cell. If intersections that connect no-pressure boundary faces are considered ($\gamma > 40°$), both models fail to predict the accurate directional permeabilities, indicating that the effect of flow dispersion within the crossing fracture may play a more important role than previously thought. However, the cumulative error boxplot in figure 6 indicates that both methods give statistically acceptable predictions of the directional permeabilities (median error of 2.7 % for the fracture-and-pipe model and a median error of 7.9 % for the fracture-only model). Thus, the systematic error observed for $\gamma > 40°$ appears negligible.

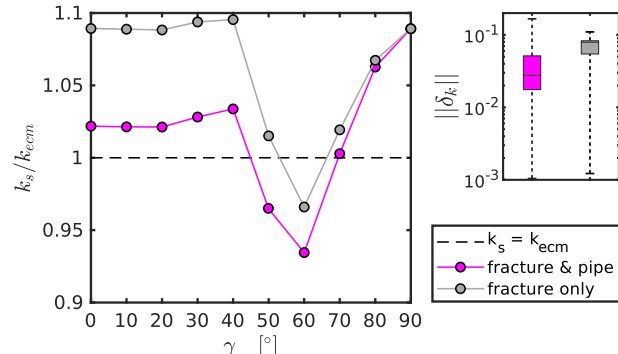

**Figure 6.** The left plot shows a comparison of directional permeabilities obtained from high-resolution Stokes-flow simulations ($k_s$) and analytically counterparts ($k_{ecm}$) derived with the ECM-approach described in the text as function of the angle $\gamma$ between the intersection and the principal flow direction. Magenta dots represent the mean permeability ratios (10 values per point) for the ECM approach described in section 4 with the half-hypotenuse pipe radius parametrization. Gray dots present the mean permeability ratio for an ECM-approach, that ignores the effect of intersections. The right plot shows a boxplot of the error norm $||\delta_k||$ computed according to eq. 15 with $k_{ecm}$ as $k_r$ for all 100 fracture-and-pipe (magenta) and fracture-only models (gray).

## 6 Intersection flow effects at network scales

In the previous section, we demonstrated the effects of IFL on the permeability of systems at the scale of a local ECM cell (i.e., system sizes near to the fractures aperture) by comparing analytically derived cell permeabilities to the results of direct-flow simulations (Stokes equations). If the intersection orientation aligns with the applied pressure gradient and connects inlet and outlet pressure faces, the permeability of the system is increased (i.e., case *b* in figure 1). To explore the effects of IFL on the permeability of systems at larger scales that cannot be fully resolved with current imaging techniques (i.e., above a few decimeters), we conduct continuum-flow simulations (as described in the appendix) of several test-cases, where IFL potentially matters. Following the results of the previous section, this should be the case for fracture formations containing two fracture sets with perpendicular strike and steep dip angles. These so-called cross-joint patterns can be naturally observed (e.g, Gross, 1993; Ruf et al., 1998; Li and Ji, 2021) and are thought to mainly result from local stress field rotations in extensional tectonic settings (Bai et al., 2002; Boersma et al., 2018). Hence, we use the software ADFNE (Alghalandis, 2017) to generate several test DFNs with two orthogonally striking (dip-directions are separated by $90°$) and vertically dipping (dip-angle of $90°$) fracture sets with constant fracture sizes for simplicity. Slight variability in dip-angle and- direction is introduced by a Fisher dispersion parameter of 20. By this, we ensured that the primary orientation of the formed intersections is oriented parallel to the z-direction in the model to provoke the possibly maximal effect of IFL on the network scale. For each test-DFN, we vary the following structural parameters during the generation process:

– The cubic overall systems side length $L$ by 1, 10, 100, and $1000\ m$.

– The constant size $l$ of all fractures in the system by 0.25, 0.5, 1, and 2 times the systems side length $L$

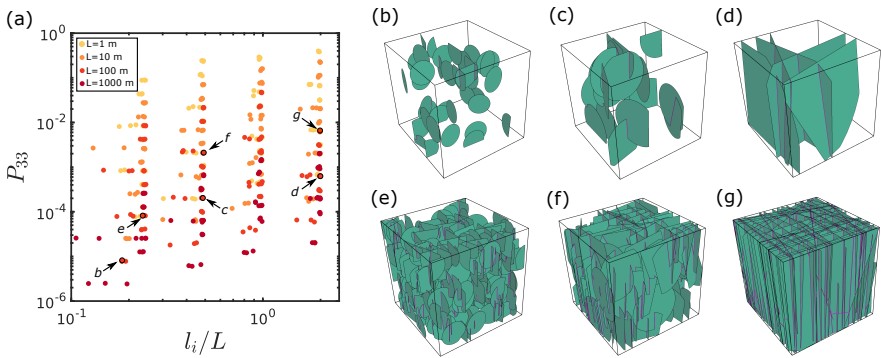

**Figure 7.** a) visualizes the geometrical configurations of all 381 test-DFNs as a function of their global $P_{33}$ value and their ratio between maximum intersection length $l_i$ to system size $L$. Colors indicate the model size $L$. Note that high global fracture porosities are predominantly achieved for smaller system sizes for the considered test cases. Panels b) to g) show the underlying DFNs structure of the models indicated in a) with system sizes of $100\,m$. Fractures are approximated with green disks and intersections with magenta lines. Panels b), c) and d) have a low prescribed number of fractures ($10*L/2l$), whereas panels e), f) and g) have a high prescribed number of fractures ($100L/2l$). The ratio of constant fracture length $l$ and system size $L$ for panels b) and e) is 0.25, 0.5 for panels c) and f), and 1 for panels d) and g).

– The total number of fractures for each set by 10, 50, and 100 times $L/2l$. The latter rescaling factor is arbitrarily chosen to increase the total number of fracture for systems with lower fracture sizes).

The following parameters are varied in the prescription of the hydraulics of each test-DFN:

– The scaling parameter $\beta$ in a sub-linear aperture-length correlation model (e.g., Olson, 2003; Klimczak et al., 2010) by $1e-4$, $1e-3$, $1e-2$. This infers the fractures hydraulic apertures $a_h$ from their sizes according to $a_h = \beta l^{0.5}$.

– The isotropic and constant permeability of the rock matrix by $1e-17$, $1e-15$, and $1e-13\,m^2$.

This results in 432 test-DFNs, which are discretized to an ECM with the two different methods already described in the
previous section to analytically derive local cell permeability tensors (i.e., the aforementioned fracture-and-pipe and fracture-only method as described in section 4).For this, we start with an ECM grid resolution of 128x128x128 numerical cells to prevent artificial connectivity (e.g., Jackson et al., 2000; Svensson, 2001) for networks with high fracture densities. If this results in a-physical fracture porosities above unity at the local cell level, we consecutively reduce the grid resolutions by powers of two up to 4x4x4 until all local cells have fracture porosities below unity. If this condition cannot be achieved (e.g.,
for small scales with high fracture densities and high apertures), the model is excluded from the analysis. Furthermore, models with hydraulic apertures above $1\,cm$ were excluded as well, as we assume that the simplification of laminar flow might not hold anymore. This results in a total of 381 test-DFNs, whose structure we quantify in a 2D non-dimensional parameter system given by (1) the ratio of the maximum intersection length of the system $l_i$ to the system size $L$ and (2) their global $P_{33}$ value (i.e., global fracture volume divided by total volume according to Dershowitz and Herda, 1992, and hence referred to as the systems

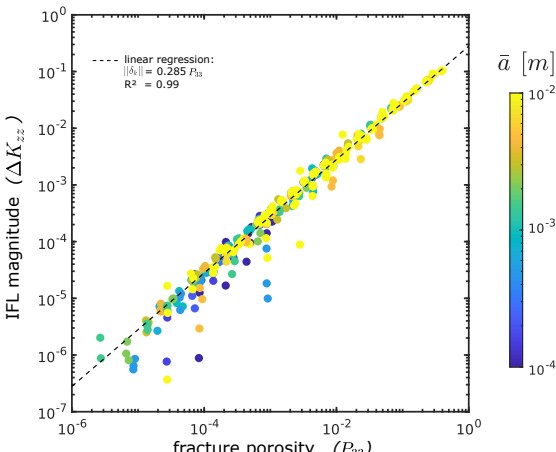

**Figure 8.** The plot demonstrates the difference between the vertical component of the permeability tensor $\Delta K_{zz}$ resulting from continuum-flow simulations of ECMs discretized with the fracture-and-pipe and fracture-only methods described in the text as a function of their respective global fracture porosities (i.e., their $P_{33}$ value). Colors indicate the constant hydraulic aperture of all fractures in the respective model. Note that high fracture densities are predominantly reached for high fracture apertures.

global fracture porosity). Figure 7 demonstrates the distribution of the generated test-DFNs within this 2D parameter space and shows the DFN structure of some chosen examples. For each geometrical DFN configuration displayed in the plot, we compute two (one for each discretization method) effective permeability tensors with the continuum flow procedure described in the appendix. We quantify the absolute difference of the vertical component in the resulting permeability tensors of the fracture-and-pipe ($k_{fp}$) to the fracture-only model ($k_{fo}$) by $\Delta K_{zz}$ according to:

$$\Delta K_{zz} = \frac{|k_{fp} - k_{fo}|}{k_{fp}}, \tag{16}$$

which serves as a measure for the magnitude of IFL effects on the network scale. Figure 8 demonstrates that the effect of IFL on network-scale flow depends linearly on global fracture porosity by $||\Delta k_{zz}|| = 0.285 P_{33}$. However, the generally very low absolute differences in vertical system permeability indicate a negligible effect of IFL at network scales. Only for networks with global fracture porosities in the range of $30 - 40\%$ we could observe differences of about $10\%$.

**7   Resolution dependency of ECM methods**

The resolution dependency of ECM methods to upscale the permeabilities of fracture networks is a crucial aspect that has to be considered to provide accurate upscaling results. Artificial connectivity is one of the main issues that arises, if the resolution of the ECM is insufficiently as demonstrated by Jackson et al. (2000) and Svensson (2001). Their results suggest that we can expect accurate upscaling results, only when the resolution of the ECM is sufficiently large to resolve the structure of

the DFN (i.e., a maximum of two fracture segments and one intersection per cell). As fracture networks typically have a multi-scale character with power-law or log-normal fracture size distributions (e.g., Bonnet et al., 2001; Davy et al., 2006), fulfilling that conditions may require very large grid resolutions. Predicting the effective permeability of the DFN by solving the groundwater flow equations (Darcy's law) would then require prior upscaling of the grid cell conductivities (e.g., Zhou et al., 2010; Hauge et al., 2012), depending on chosen flow solver and the available computational resources. However, averaging

or flow-based upscaling approaches may misrepresent network-scale flow characteristics, depending on the chosen coarse grid resolution. Hence, it is often unclear how the resolution dependency affects the accuracy of effective permeability computations and whether flow anisotropy is conserved. In the following, we will demonstrate that using ECMs of DFNs with sufficiently high resolutions is capable of doing this while avoiding initial upscaling. For this, we compare effective permeability tensors obtained from massively parallelized continuum flow simulations (see Appendix A) for different DFN scenarios with varying

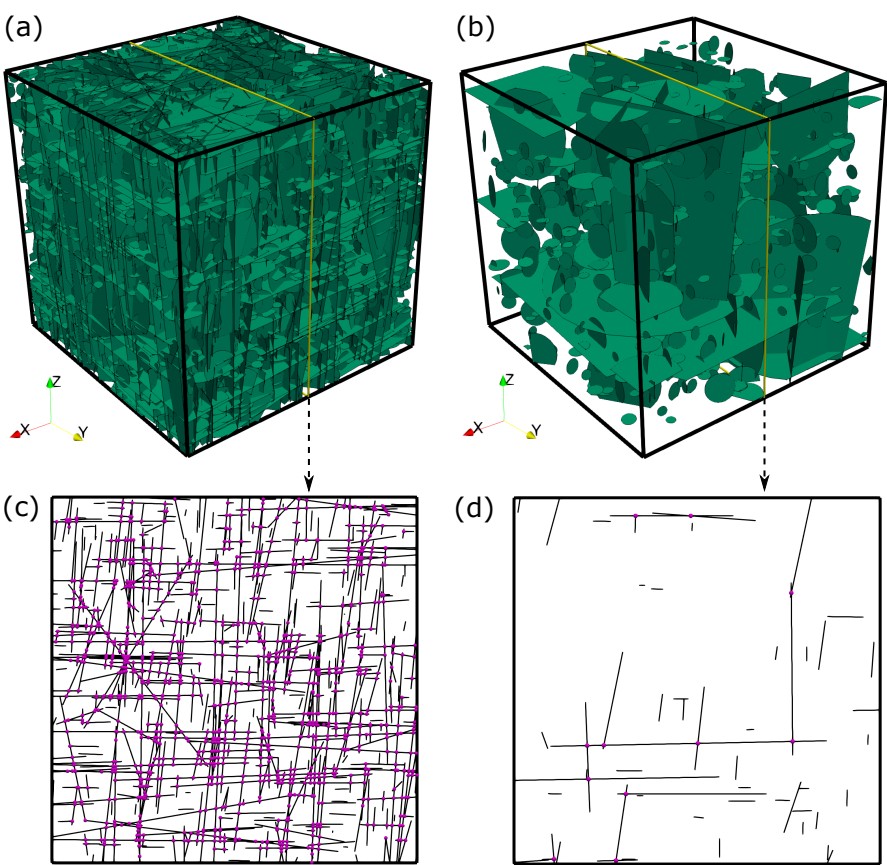

**Figure 9.** a) and b) display the test DFNs with 10000 and 1000 fractures, respectively. Both are generated with the software ADFNE (Alghalandis, 2017), whereas input parameters are given in the text. Yellow lines depict the location of the slice shown in c) and d). There, black lines indicate fractures and magenta spheres the location of fracture intersections.

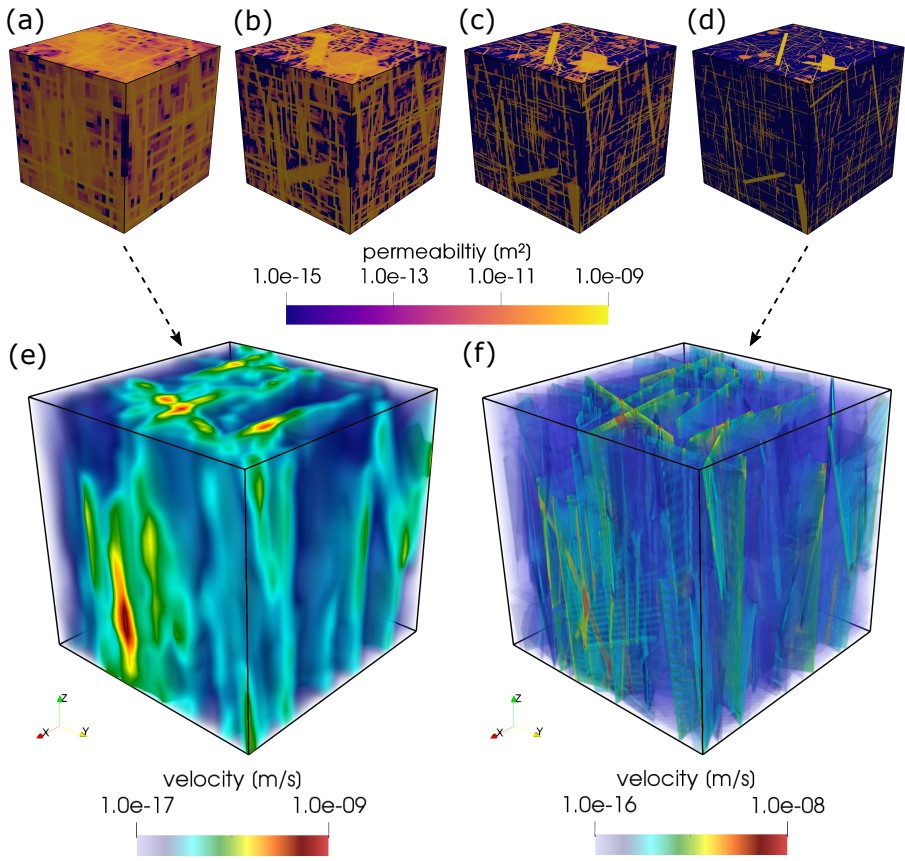

**Figure 10.** a), b), c) and d) display the norm of the permeability tensor for each cell in an ECM representations of the 10000 fracture test DFN displayed in figure 9 a) for grid resolutions of $32^3$, $64^3$, $128^3$ and $256^3$ voxels, respectively. e) and f) visualize the resulting velocity distribution for an applied pressure gradient in z-direction.

resolutions of their equivalent continuum counterparts. We generate two test DFNs utilising the open-source MATLAB toolbox ADFNE (Alghalandis, 2017). For comparability reasons, we use similar input data as Hadgu et al. (2017), who separated all fractures into three orthogonal sets, based on the data reported in SKB (2010). $S1: 90|090$, $S2: 90|000$, $S3: 00|360$ give the mean dip-angle and dip-direction for the three fracture sets, respectively with a constant Fisher distribution concentration value of 5 accounting for variability around the mean. Fracture sizes $l$ are distributed as a power law according to:

$$l = \left[ \left( l_1^{\alpha+1} - l_0^{\alpha+1} \right) u + l_0^{\alpha+1} \right]^{1/\alpha+1}, \tag{17}$$

whereas $l_1$ is the upper cut-off length ($500\,m$) and $l_0$ the lower cut-off length ($15\,m$), $u$ represents a set of uniformly distributed random numbers in the interval $(0, 1)$ and $\alpha$ the power law exponent (here $\alpha = -2.5$). All fracture centers are randomly placed in a cube with $500\,m$ side lengths (the resulting DFNs are displayed in figure 9) with a background matrix permeability of $10^{-18}\,m^2$. A sub-linear scaling of aperture versus length (e.g., Olson, 2003; Klimczak et al., 2010) is employed to correlate

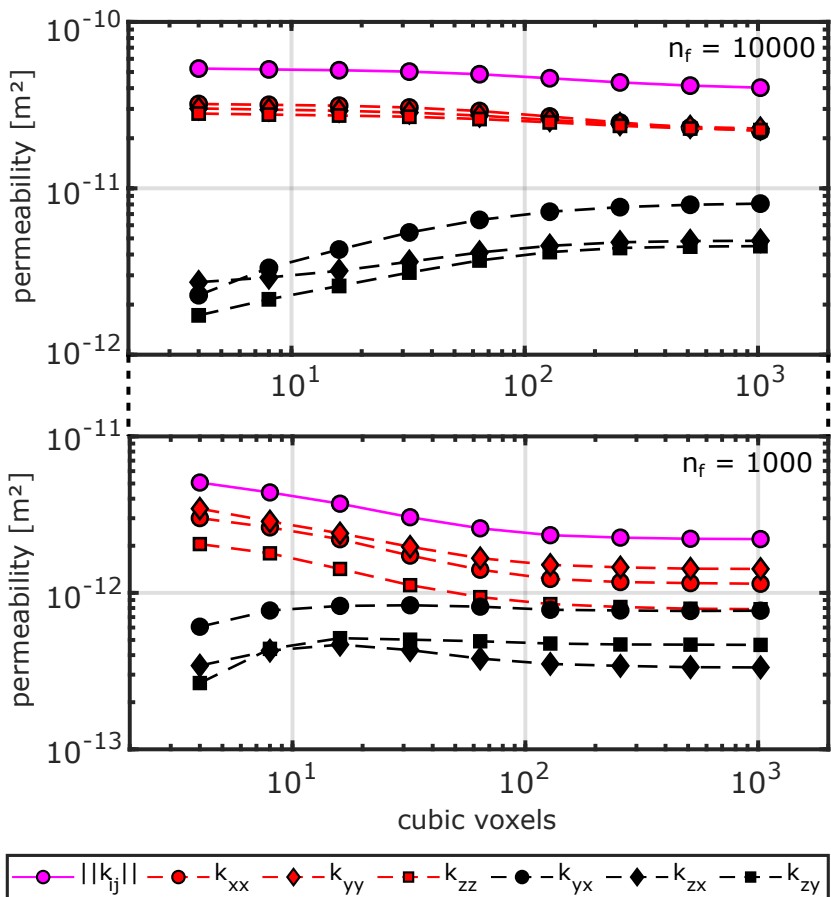

**Figure 11.** Absolute permeability values $k$ for the 6 main components of the computed effective permeability tensor (principial components in red, off-diagonal components in black) and the norm of the permeability tensor in magenta as a function of the grid resolution in cubic voxels (number of voxels in x-y-z direction).

the hydraulic apertures $a_h$ of the fractures to their lengths $l$:

$$a_h = \beta \, l^{0.5}, \tag{18}$$

with a scaling factor $\beta$ of $10^{-4}$. The only difference between the two test DFNs is the overall fracture number, which is 10000 for the DFN-A (plot a in figure 9) and 1000 for the DFN-B (b in figure 9), such that we obtain a densely and sparsely fractured system, respectively. DFN-A thus represents the scenario of a typical REV network, according to Long et al. (1982);

Oda (1985). DFN-B, on the other hand, reflects a flow scenario closer to the percolation threshold with anisotropic, non-REV behaviour (Maillot et al., 2016).

After calculating all fracture intersections with ADFNE's built-in function Intersect (see b and d in figure 9 for intersection spots in a 2D slice), we use the method presented in section 4, which incorporates the permeability parametrization concepts

from section 3, to generate several ECMs with varying grid resolutions. Starting from $4^3$ voxels and increasing by powers of two up to $1024^3$ voxels yields 9 different continuum representations for each test DFN (see figure 10 for examples). For each representation, we compute the effective permeability tensor of the DFN by repeatedly solving the Darcy equations in three principal flow directions (see Appendix A for a detailed description). The results are displayed in figure 11. For both test DFNs, the norm of the resulting effective permeability tensor ranges within the same order of magnitude. For DFN-A, we obtain a difference of about 30 % from coarse ($4^3$ voxels, $||k_{ij}|| = 5.24 * 10^{-11}$) to fine ($1024^3$ voxels, $||k_{ij}|| = 4.03 * 10^{-11}$) grid resolution, whereas DFN-B shows a larger difference of about 129 % (coarse $||k_{ij}|| = 5.07 \cdot 10^{-12}$, fine $||k_{ij}|| = 2.21 \cdot 10^{-12}$). Thus, the resolution dependence of the absolute permeability is small for fracture networks with an expected REV behavior (DFN-A) and more pronounced if fracture networks with non-REV behavior (DFN-B) are considered. Interestingly, the individual components of the permeability tensor converge to constant values above resolutions of $128^3$ voxels for both test cases, indicating that anisotropy magnitude depends on the level of detail of the ECM grid.

## 8   Discussion

Including a pipe-flow model into the ECM generation process improves the representation of permeability anisotropy therein and can have impacts on overall permeabilities as well. For example, at the scale of the intersection itself, it significantly modifies the shape and absolute values of the permeability tensor (figure 4). However, the errors of the intersection benchmark (2.7 % and 7.9 % for the fracture-and-pipe and fracture-only model, respectively) indicate that, from a statistical perspective, the effect of IFL on overall permeability is usually a second-order effect. This is because the fracture-only ECM discretization approach by default accounts for the increased permeability of intersection cells in an isotropic manner, simply by the summation of the individual fracture contributions per cell. The parameter study from section 6 shows that incorporating the effects of IFL into an ECM-based upscaling approach is only necessary for fracture systems that (1) produce a significant amount of fracture intersection (i.e., for fracture systems with two fracture sets of strongly different directions) and (2) have sufficiently high global fracture porosities (above 30 %), regardless of scale. Achieving the latter in our test-scenarios is strongly coupled to large fracture apertures (in the order of $1\ cm$, see figure 8) and a ratio between mean intersection and system size close to or above unity (i.e., intersections fully penetrating the system). So, if two dominant orthogonal fractures with large apertures form an intersection that is penetrating the whole system along the direction of flow, the effect of IFL influences the systems effective permeability. Due to the non-linear radius-permeability relation, this may become more important for fractures with apertures above $1\ cm$. However, for apertures above $1\ cm$, Reynolds numbers can easily exceed the critical threshold of unity (e.g., Zimmerman and Bodvarsson, 1996), which would require non-linear concepts to relate fracture and intersection permeability (e.g., Forchheimers law), as well as Navier-Stokes rather than Stokes simulations.

If small DFNs with sizes closer to the mean hydraulic radius of the intersections (e.g., micro-fracture networks of $1\ m$ size, that cannot be naturally resolved with current imaging techniques) are considered for permeability prediction, IFL could play an important role. Then, however, additional factors have to be considered as well. For example, de Dreuzy et al. (2012) have shown that fracture scale heterogeneity affects network scale connectivity due to flow channeling caused by closure in the

aperture field. This may appear if the ECM cell size is similar to the internal correlation length of the fractures (e.g., Méheust and Schmittbuhl, 2003; Kottwitz et al., 2020) which would ultimately require new concepts to account for deviations from the average flow behavior instead of using fracture permeability parametrizations. A possible solution would be to introduce frac-

ture permeability fluctuations if the ECM cell size is smaller than the individual fractures correlation length. Unfortunately, the scaling of the correlation length in fractures is poorly understood, so further research is required before integrating these effects. Additionally, the pipe parametrization we use as a first-order approximation for intersection permeability requires refinement to account for irregular shapes, tortuosity, or closure, representing another interesting question to solve in future studies.

For flow simulations at reservoir scales (similar to the test cases considered in section 7), the only computationally feasible

solution is to use parametrization concepts (e.g., section 3). For that, we were able to demonstrate that the presented fracture-and-pipe ECM method is capable of providing converged effective permeability tensors if the ECM resolution, i.e., the ratio of system size to discretization step size, is sufficiently large. This resolution dependency for 3D ECMs has not been reported at this level of detail so far but was expected based on previous works of Jackson et al. (2000) and Svensson (2001). There, the main problem is identified as artificially increased connectivity at lower resolutions, which occurs if the resolution is larger

than either the average spacing of the fracture network or the minimal fracture length of the DFN, leading to overestimated permeabilities and misinterpreted anisotropy. Here, we use the average minimal distance of each fracture center to all other fracture centers in the network as a first-order approximation for fracture spacing. With an average spacing of $13.1 \pm 4.5\ m$, continuum grid resolutions above roughly 38 cubic voxels should theoretically start preventing artificial connectivity for DFN-A. For DFN-B with an approximated average spacing of $28.9 \pm 10.9\ m$, the required resolution to damp that effect is even lower

(about 17 cubic voxels). Both test DFNs have the same lower cutoff fracture size of $15\ m$, so artificial connectivity should start decreasing above resolutions of about 33 cubic voxels. Looking at figure 11, we observe ongoing permeability convergence at these three mentioned resolutions. We attribute this to the fact that fractures are spaced randomly in space but sampled with a regular grid. Thus, the distance between fracture tips and continuum cell-edges might be larger for low resolutions, again causing permeability overestimations. Only above a resolution of 128 cubic voxels, all these effects seem to dampen out,

allowing to declare the solution as sufficiently converged with quantitative errors below $10\%$ for tensor norm and individual components. Hence, we suggest a general upper boundary of a third of the minimal fracture length $l_0$ as cell size for an ECM discretization of a DFN to provide constant results.

Based on analytical solutions of flow in fracture networks with constant apertures, Svensson (2001) proposed that the ratio of ECM cell size to hydraulic aperture should not exceed two to provide small flow errors. So far, the ratio of cell size to the

minimal hydraulic aperture in the system was much larger (about 1260) due to the low scaling factor $\beta$ of the sub-linear aperture to length correlation (eq. 18). To achieve similar discretization ratios of Svensson (2001) while maintaining a power-law size scaling, we would have to increase $\beta$ to $10^{-1}$, resulting in minimal and maximal apertures of $0.39$ and $2.14\ m$, respectively. As this would violate the assumption of laminar flow conditions within the fractures, we cannot test their hypothesis and rather recommend staying above the maximum hydraulic aperture $a_{h1}$ of the system, as otherwise the volume-fraction

based permeability scaling factor in equations 13 and 14 exceed unity. In that case, parametrization assumptions might not hold anymore, preventing the use of continuum flow methods. However, as demonstrated here, sticking to $l_0/3 > \delta x > a_{h1}$ as

condition for ECM discretization delivers constant effective permeabilities and conserves flow anisotropy for the upscaling. Within that discretization range, mapping a DFN onto an equivalent continuum grid can be used as a geometric upscaling procedure for further effective permeability analysis. Notably, this range strongly depends on the structural characteristics of the considered DFN, especially on the fracture size distribution and corresponding aperture correlation functions. For some DFNs this may require to crop the fracture size distributions from below to a few multiples of the cell size and compensate the hydraulic contribution of lower sized fractures with a background permeability.

## 9    Conclusions

This study analyzed the complexity of fracture intersection flow by conducting Stokes-flow simulations in simple fracture crossings. Intersections that are aligned with the pressure gradient initiating the flow cause an increase in permeability, as they act similar to a pipe. This results in intersection flow localization (IFL), i. e., intersections represent preferred pathways for the fluids compared to the connected fractures. We thus extended the state-of-the-art methodology to generate equivalent continuum models (ECM) for effective permeability computations of discrete fracture networks (DFN) to incorporate IFL effects. Those are integrated using a directional pipe-flow parametrization with a hydraulic radius of half the hypotenuse size in a right-angled triangle with side lengths of both intersecting hydraulic apertures. By assessing the permeabilities of fracture intersections numerically, we could demonstrate that for system sizes close to the approximated pipe radius ($mm$ to $cm$), the effect of IFL on permeability can be almost one order of magnitude. At network scales ($m$ to $km$), the impact of IFL on the systems effective permeability is generally minor. Only for fracture systems with high global fracture porosities (above $30\%$) IFL effects become noticeable. Analyzing the effects of IFL on mass transport through fracture networks poses an interesting question for a follow-up study. For example, Makedonska et al. (2016) have shown, that early breakthrough times of solute transport through kilometer-scale DFNs are sensitive to local permeability fluctuations. Thus, local permeability increases induced by IFL could potentially affect transport behavior as well.

Next to the effects of IFL, we investigated the resolution dependency of current ECM-based upscaling approaches, as the cell size with which the ECM is discretized represents the most crucial aspect for the accuracy of ECM-based effective permeability predictions. Based on a resolution test with two different DFN scenarios, we suggest that the ECM cell size should be smaller than a third of the minimal fracture size and larger than the maximal hydraulic aperture of the system to conserve constant permeabilities and full anisotropy of flow. Within that range, we conclude that ECM methods equivalently serve as geometric upscaling procedures for fluid flow problems. It is important to note, that the accuracy of ECM methods to predict flow are always linked to the quality of the input DFN. Improving the DFN method to better characterize natural fracture systems, especially in terms of fracture termination rules and spatial clustering, is still an ongoing topic of research.

## Appendix A: ECM-based effective permeability prediction workflow

In the following, we will explain our method to obtain the effective permeability tensor of continuum cell representations for fractured-porous media. The governing equations for steady-state single-phase flow equations for an incompressible, isothermal and isoviscous fluid without sources and sinks are given im compact form by the following system of mass (eq. A1) and momentum (eq. A2) conservation equations:

$$\nabla \cdot q = 0, \tag{A1}$$

$$q = -K\nabla P, \tag{A2}$$

whereas $\nabla$ and $\nabla\cdot$ denote the gradient and divergence operator for global 3D Cartesian coordinates, respectively. The specific discharge (flux) is given by $q$, pressure by $P$ and the positive definite and symmetric hydraulic conductivity tensor by $K$ according to:

$$K = \begin{bmatrix} k_{xx} & k_{yx} & k_{zx} \\ k_{yx} & k_{yy} & k_{zy} \\ k_{zx} & k_{zy} & k_{zz} \end{bmatrix} \frac{\rho g}{\mu}, \tag{A3}$$

with the principal permeability tensor components $k_{xx}$, $k_{yy}$ and $k_{zz}$, the off-diagonal components $k_{yx}$, $k_{zx}$ and $k_{zy}$ as well as fluid density $\rho$, gravitational acceleration $g$ and fluid dynamic viscosity $\mu$. We employ a 3D finite-element discretization scheme (e.g., Hughes, 1987; Zienkiewicz and Taylor, 2000; Belytschko et al., 2000; Lin et al., 2014) for equations A2 and A1 to simulate boundary driven pressure diffusion through any input grid consisting of unique permeability tensors. Using the Galerkin method (e.g., Belytschko et al., 2000; Lin et al., 2014), we transform equation A1 into an expression for the nodal residual $R$ according to:

$$R = \int_V \nabla N^T K \nabla N dV P = 0. \tag{A4}$$

$V$ denotes the domain volume, $N$ the nodal shape function matrix and $P$ the nodal pressure. We use 8-node rectangular elements (voxels) with linear interpolation functions (e.g., Zienkiewicz and Taylor, 2000) for volume integral approximation, whereas element integrals are evaluated by Gauss-Legendre quadrature rule (e.g., Belytschko et al., 2000) over 8 integration points with parametric coordinates. Within each element, standard coordinate transformation is employed to compute shape function derivatives with respect to global coordinates $\nabla N$:

$$\nabla N = J^{-1}\nabla_L N, \quad J = \nabla_L N x, \tag{A5}$$

where $\nabla_L$ denotes gradient operator for local 3D element coordinates, $J$ the Jacobian matrix and $x$ the 3D global element coordinates. After imposing initial pressure conditions at the boundary nodes, the global residual vector $R_g$ is assembled from

elemental contributions (e.g., Hughes, 1987) according to eq. A4 to solve the linear system of equations:

$$C_g P^{new} = R_g, \tag{A6}$$

for the unknown pressure $P^{new}$. $C_g$ denotes the global coefficient matrix, which is assembled from the nodal coefficient matrix $C$ given by:

$$C = \int_V \nabla N^T K \nabla N dV. \tag{A7}$$

Following this, we evaluate the Darcy velocities at the integration points $u$ based on the newly solved nodal pressures by:

$$u = K \nabla N P^{new}, \tag{A8}$$

whereas the velocity vectors on the nodes are averaged from the neighboring integration points.

Three principal directions of the applied pressure gradient have to be considered to predict the full tensor of permeability. Thus, the flow simulation procedure has to be repeated three times such that each principal flow direction ($x$-, $y$- and $z$-direction in a Cartesian coordinate system) is covered. For each iteration, two constant pressure values are applied at two opposing boundary faces (e.g., lower and upper face in a cube for principal flow in z-direction) and the same linear interpolation between those two values is applied at the remaining four boundary faces (see figure A1 for an example). This ensures to capture both, the diagonal and off-diagonal terms of the permeability tensor properly, which are computed by substituting the volume average $\bar{u}$ of all nodal velocity vectors $u_I$ (see eq. 3) into Darcy's law for flow through porous media in the form of eq. 4. Figure A1 displays the situation of a vertically aligned pressure gradient ($\Delta P_z = \frac{\delta P}{\delta z}$). The corresponding entries in the permeability tensor are computed according to:

$$\begin{bmatrix} k_{zx} \\ k_{zy} \\ k_{zz} \end{bmatrix} = \frac{\mu}{\Delta P_z} \begin{bmatrix} \bar{u}_x \\ \bar{u}_y \\ \bar{u}_z \end{bmatrix}, \tag{A9}$$

and vice versa for the iterations with pressure gradients in $x$- and $y$- direction to obtain the permeability tensor as shown in eq. A3.

The used single-continuum discretization scheme might appear simplistic compared to more sophisticated mesh-representations (see Berre et al., 2019). However, the merits of our approach rather lay (1) on a fully anisotropic permeability representation of the individual continuum cells and (2) massive parallelization and HPC optimization. Utilizing the parallelization framework of PETSc (Balay et al., 2018) and their multigrid preconditioned solvers significantly reduces the computational cost, allowing simulations routinely with $10^9$ individual grid cells. An increase in grid resolution compensates for the benefits of using conforming meshes or multi-continuum formulations (e.g., Berre et al., 2019). To test this, we compare our modeling procedure against benchmark case 1 from Berre et al. (2020), who compare 17 different methods of simulating single-phase flow in fractured porous media. The initial setup (displayed in a) in figure A2) consists of an inclined fracture with a hydraulic

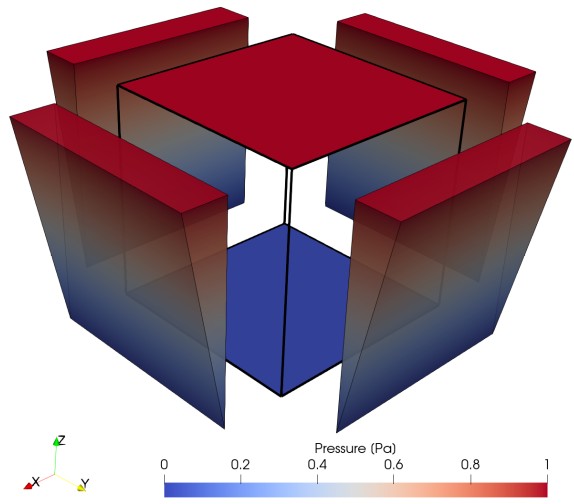

**Figure A1.** Pressure boundary conditions for an applied gradient in $z$-direction. Here, top and bottom faces experience constant pressures of 1 and 0 $Pa$, respectively. A linearly interpolated pressure distribution is applied at the remaining four boundary faces, as indicated by the coloured wedges next to the side-faces of the model. Thus, the principal direction of flow is in $z$-direction, allowing to calculate the $z$-component related terms of the permeability tensor according to eq. A9

aperture of $10^{-2}$ $m$ embedded in a cube of $100$ $m$ length with a matrix hydraulic conductivity of $10^{-6}$ $m^2$, whereas the hydraulic conductivity of a small band of $10$ $m$ width at the bottom is increased to $10^{-5}$ $m^2$. We prescribe these two values as background permeabilities and use the methodology described in section 3 to incorporate fracture permeability accordingly. The boundary conditions are given by small pressure inlet ($4$ $Pa$) and outlet ($1$ $Pa$) bands as indicated in plot a in figure A2. The comparison of the pressure distribution (plot b in figure A2) highlights that already with a resolution of $32^3$ voxels, we obtain a good fit with the benchmark target field. This thus suggests that our modeling procedure is sufficiently correct for effective permeability predictions.

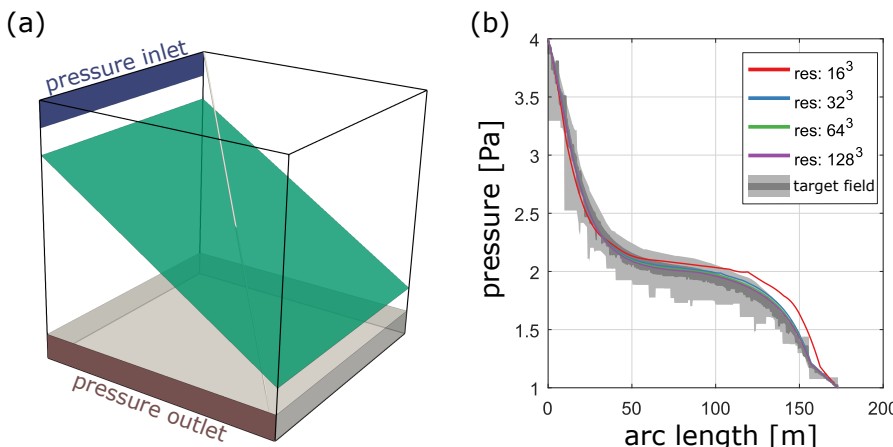

**Figure A2.** Benchmark case 1 from Berre et al. (2020). a) shows the benchmark geometry of an embedded fracture (aperture of $10^{-2}$ $m$) in a matrix with a hydraulic conductivity of $10^{-6}$. The hydraulic conductivity in the grey band at the bottom is increased to $10^{-6}$. Constant pressures of $4$ $Pa$ and $1$ $Pa$ are applied at the inlet band (blue) and outlet band (red), respectively. The diagonal light grey line through the model indicates the sampling line for the pressures shown plot in b). There, the pressure distribution is plotted as a function of arc length of the gray line in a), and the results of different resolutions are compared to the benchmark target field obtained from 17 different numerical methods. The dark grey region illustrates the area between the 10th and 90th percentiles for the highest refinement level of the benchmarked methods, whereas the light grey region illustrates the same area for their lowest refinement level.

*Code and data availability.* FD Stokes-Flow: https://bitbucket.org/bkaus/lamem/src/master/ ; commit: 9c06e4077439b5492d49d03c27d3a1a5f9b65d32

FE Darcy-Flow: https://bitbucket.org/mkottwitz/anisotropicdarcyupscaling/src/master/ ; commit: feaed524bcef636725ee30b2d4d3136b6525d83e

The scripts to reproduce the input data sets are available upon request from the authors.

*Author contributions.* MOK wrote the initial draft of the manuscript, performed numerical simulations, analysed the data and generated the figures. AAP supervised and helped designing the study, helped implementing the numerical methods and edited the manuscript. SA helped

designing the study and edited the manuscript. BJPK edited the manuscript and discussed the results.

*Competing interests.* The authors declare that they have no competing interests

*Acknowledgements.* This work has been funded by the Federal Ministry of Education and Research (BMBF) program GEO:N, grant no. 03G0865A and the M3ODEL consortium at Johannes Gutenberg-University Mainz. The authors gratefully acknowledge the computing time granted on the supercomputer Mogon II at Johannes Gutenberg University Mainz (hpc.uni-mainz.de). The authors sincerely thank two

anonymous referees for reviewing this manuscript.

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
