# Peer review of "Investigating the effects of intersection flow localization in equivalent continuum-based upscaling of flow in discrete fracture networks"

_Solid Earth, 2020_

## Author Comment (AC1)

We sincerely thank the anonymous referee for reviewing this manuscript. The constructive comments helped us to clarify the scope of our manuscript. Please find below a point-by-point response to the referees' comments (comments of the reviewer in black and our response in red)

On behalf of all authors,

Yours sincerely,

Maximilian Oskar Kottwitz

**Scope and implications:**
The one major comment I have is about scope. The authors very clearly lay out their theoretical and numerical arguments. What I was missing is a bit more on the scope and on how useful their methodology is for characterizing natural systems. Such a discussion should include a short review of how well and on which scales fracture networks can be characterized in the first place (e.g by geophysical methods and/or imaging techniques).
I am mainly thinking about scales here. My understanding is that if samples can be CT-scanned (e.g. mm to cm-scales) direct flow simulations that resolve the actual fracture geometries would be performed without simplifying the fractures as simple geometric entities. Here flow at fracture intersections would be naturally resolved. On larger scales, the apertures of fractures are very hard to determine and consequently fractures are often represented in models as reduced-order elements – the DFN approach. It remained a bit unclear to me for which input datasets and flow simulation approaches, flow localization at fracture intersection really needs to be considered in the way the authors describe. The authors address these points in the discussion and conclusion sections, where they talk about system sizes but I think it would help the reader if the authors expanded this discussion.

We fully agree on the necessity to clarify the scope of our presented ECM approach and the scale dependence of IFL effects.
Generally, the structure of fractured systems is intrinsically multi-scale and thus requires different modeling approaches, broadly separated in direct- and continuum-flow approaches. Direct flow simulations (using Navier-Stokes or Stokes equations) require an explicit representation of a medium's void space, which is only naturally retrievable by non-destructive imaging techniques (CT-scans, for example). However, they are limited to mm- or cm-scales, as you pointed out. Above these scales, pore-spaces cannot be imaged anymore (so natural data are not available) and direct-flow simulations become computationally infeasible to conduct. To be still able to simulate flow at scales above a few cm, we thus have to either use the DFN approach (if matrix is impermeable) or continuum-flow approaches. They average the hydraulics of any rock medium to an effective permeability tensor on the local scale (computational cell, size above 1 centimeter) to simulate flow on the global scale (meters to kilometers). So the ECM method used in this study only applies to these "continuum-scales" at which no direct representation of the mediums void space is possible.
To clarify this in the paper, we extend and partly rephrased the introduction from lines 28 to 33 according to:

*"Numerical modeling of fluid flow is most accurately based on the Navier-Stokes equations (Bear, 1972). For a single phase of incompressible and iso-viscous fluid in an iso-thermal system, they simplify to the Stokes equations if laminar flow conditions are considered (i.e., Reynolds numbers below 1 -10). Assuming an impermeable rocks matrix, one can solve for the velocity distribution resulting from prescribed pressure boundary conditions, allowing to determine the rocks effective permeability utilizing Darcy's law for flow through porous media (e.g., Andrä et al., 2013b; Osorno et al., 2015; Eichheimer et al.,2019, 2020; Kottwitz et al., 2020). Those so-called direct-flow modeling approaches crucially rely on a digital*

*representation of a rock that separates pore-space from the matrix, which results from high-resolution X-ray computed tomographies (Andrä et al., 2013a; Cnudde and Boone, 2013). However, they are limited in maximum scannable size and respective trade-off to numerical resolution, making them applicable to small scales only (nanometers to a couple of centimeters at most). At larger scales (above a couple of centimeters), so-called continuum-flow approaches serve to model fluid flow, usually based on the concepts for flow through porous media proposed by Darcy (Darcy, 1856). Instead of a representation of the medium's pore-space, they require an initial hydraulic representation of the medium. This is given by prescribed effective permeabilities for certain control volumes within the medium, which upscale hydraulic properties from smaller scales to observation scales. Thus, the key of this so-called upscaling problem (e.g., Zhou et al., 2010; Hauge et al., 2012; Lie, 2019) is to adequately represent the rock structure with an appropriate model of effective permeabilities, which for fractured rock masses is often cumbersome due to their structural heterogeneity (Dershowitz and Einstein, 1988; Odling et al., 1999). The main problem is that acquiring detailed natural fracture data in 3D is intricate, as seismic imaging techniques suffer from resolution limits (Cartwright and Huuse, 2005; Malehmir et al., 2017), preventing a multi-scale structural assessment of individual features in fracture formations. Hence, outcrop (2D) and borehole (1D) studies are the only possibilities to acquire detailed natural fracture data, despite their reduced dimensionality (Lei et al., 2017), and acquiring deterministic knowledge of all individual structures in a fracture formation is impossible. Due to this, the discrete fracture network (DFN) method has been extensively used as a conceptual framework to provide statistically-based approximations of real fracture networks for decades (Long et al., 1982; Cacas et al., 1990; Bogdanov et al., 2003; Darcel et al., 2003; Xu and Dowd, 2010; Davy et al., 2013; Maillot et al., 2016). In this approach, each fracture in a given network is represented with a reduced order object (lines in 2D and discs or rectangles in 3D) with a prescribed location, size, and orientation. Naturally measured structural properties like size- and orientation-distributions (Odling et al., 1999; Healy et al., 2017), as well as fracture density and spacing (Ortega et al., 2006), serve as a quantitative basis to prescribe their geometrical properties (e.g., Hyman et al., 2015; Alghalandis, 2017)."*

In this paper, we demonstrated and quantified the effects of IFL on the permeability tensor at the local scale by conducting direct-flow simulations at scales where it is possible to conduct those (1cm system size) while maintaining a sufficiently high numerical resolution to ensure the accuracy of the results. Based on these results, we demonstrated that a fracture-and-pipe parametrization as presented here results in a more accurate representation of the local scale hydraulics than a fracture-only parametrization (usually considered in comparable studies from the literature). Admittedly, we only briefly discussed the importance of IFL effects at the global scale, where we predominantly focused on the resolution dependency of current ECM methods. We thus performed a more sophisticated parameter study of IFL effects on network-scale permeability to extend the discussion part of this study. As this was also the main request of the second reviewer, we would refer to our reply to the review of the second reviewer at this stage.

Another point that the authors may want to discuss is if their effective permeability models also preserve other properties, like e.g. break-through times, spatial pressure variations, and solute transport pattern. It's a bit outside the scope of the paper, so the authors do not need to do this – I just kept thinking that break-through times would probably be affected for fracture networks where ILF matters.

This indeed represents an exciting question as solute or particle transport tend be sensitive to local permeability variations. For example, Makedonska et al. 2016 (DOI:

10.1016/j.advwatres.2016.06.010) have shown that early breakthrough times are sensitive to local changes of permeability, induced by in-fracture aperture variability. However, addressing this issue in more detail requires the solution of the transport problem, and this is – as you already pointed out – outside the scope of this paper but could be the scope of a follow-up study. We added the following statement to the conclusion part at Line 351:

*"Analyzing the effects of IFL on mass transport through fracture networks poses an interesting question for a follow-up study. For example, Makdonska et al., (2016) have shown, that early breakthrough times of solute transport through kilometer-scale DFNs are sensitive to local permeability fluctuations. Thus, local permeability increases induced by IFL could potentially affect transport behavior as well."*

**Minor technical points:**

l. 7: It's not really a problem to include matrix properties in full Stokes simulations. All major CFD packages involve multi-physics solvers that can handle energy or mass exchange between solid and fluid regions. Maybe unnecessary to make this statement here?

In full Stokes simulations the viscosity of the matrix is significantly larger than that of the fluid (usually water) which is why it makes sense to assume the matrix to be rigid and only simulate the motion of fluid through the pores/cracks. However, we are aware that the major commercial CFD packages indeed have the possibility to include matrix properties (by using hybrid formulations based on the Stokes-Brinkman eq. for example). The situation is different when performing Darcy-like continuum-flow simulations in fractured reservoirs, where the matrix can indeed be incorporated in a straightforward manner. Including such matrix properties in fracture network simulations is actually one of the significant advantages of continuum flow methods (e.q., ECM method) compared to discrete flow methods in fracture networks (e.g., DFN method), which assume the matrix to be impermeable.
We rephrased the abstract starting at line 7 to:

*"While continuum methods have the advantage of lower computational costs and the possibility of including matrix properties, choosing the right cell size to discretize the fracture network into an ECM is crucial to provide accurate flow results and conserve anisotropic flow properties."*

l. 12: I assume the authors coined the term "intersection flow localizations (IFL)" if so, please make it clear that this is your invention and not something that's established in the literature.

Yes, we haven't found it elsewhere. Thus, we changed line 12 to:

*"… in a process, we term intersection flow localization (IFL)."*

l. 28f: Maybe expand this, to make it clear how natural fracture networks can be characterized?

See our answer to the first reviewer comment, where we indicated an extension of the introduction regarding this and multi-scale modeling approaches.

l. 64: Are you sure pflotran and modflow have these shortcomings?

Modflow was recently extended with the "XT3D" option in their Node Property Flow Package (Modflow 6, see https://pubs.er.usgs.gov/publication/tm6A56). By that, they enable

incorporating a full three-dimensional permeability tensor into the simulations at the cost of an increased local stencil in their staggered-grid finite-difference discretization scheme. However, up to now this still runs sequentially and not in parallel, making it difficult to conduct high-resolution simulations.

Pflotran on the other hand is massively parallelized by utilizing the PETSC interface to MPI, but until now doesn't have the possibility to include permeability tensors at the local level. Hence the flow-solver we developed combines the advantages of both codes. We rephrased line 64 accordingly:

*"There, current issues in commonly used 3D flow solvers, such as PFLOTRAN (Lichtner et al., 2016) are a lack of a fully anisotropic permeability representation at the local cell level. So-called stair-case patterns … predicting effective permeabilities of fractured media. On the other hand, MODFLOW (McDonald and Harbaugh, 1988) introduced support for local permeability anisotropy but not within a massively parallelized framework, making it difficult to conduct large numbers of high-resolution simulations. However, assessing permeabilities in a Monte-Carlo-like framework (e.g., de Dreuzy, 2012) is necessary to explore the variance of hydraulic system properties induced by stochastically generated input data. Hence, a flow-solver that combines the advantages of local permeability anisotropy and massive parallelization should be beneficial for numerical permeability assessments of fracture networks."*

Equation 4: Shouldn't there be a length scale over which the pressure drop deltaP occurs?

Correct – thanks for pointing this out. Accidentally, we confused the delta sign with the gradient operator, which intrinsically incorporates length scale measures. We added length-scale parameter (*L*) accordingly in equations 4,5,6 and 8.

l. 256: Maybe expand what's in this SKB 2010 reference? This is the only place where you talk about naturally systems; I think it would help to be more specific.

The two test cases we generated to demonstrate the resolution dependency of ECM upscaling methods could have been chosen completely arbitrarily, as our main intention was to put the ECM method at test and not targeted to characterize peculiar natural systems. Discussing the applicability of the DFN method to characterize natural systems is, of course, a highly relevant question (issues of fracture terminations or spatial clustering, for example) and still an ongoing research topic, but out of the scope of this study. The accuracy of the ECM method to predict network permeabilities is strongly depending on the quality of the input DFN, which itself is always a question of available data and resources to acquire this data. To clarify this, we added the following line to the conclusion at line 356:

*"It is important to note, that the accuracy of ECM methods to predict flow are always linked to the quality of the input DFN. Improving the DFN method to better characterize natural fracture systems, especially in terms of fracture termination rules and spatial clustering, is still an ongoing topic of research."*

Only for the sake of comparability we have chosen to use similar input data as provided by the DFN/ECM comparison study of Hadgu et al. (2017). Their input data were based on measurements reported for the cited SKB project, but strictly speaking generic, as they slightly manipulated the data (as written in table 1 of their study). We thus rephrased line 256 to:

*"For comparability reasons, we use similar input data as Hadgu et al.(2017), who separated all fractures into three orthogonal sets, based on the data reported in SKB (2010)."*

l. 390: are these really nodal velocities? Or rather cell (integration point) velocities (as stated in the next line)?

> You are right, these are actually velocities in the integration points. The nodal Darcy velocities are then averaged from the surrounding integration point velocities.  We changed line 389 accordingly:
> *"Following this, we evaluate the Darcy velocities at the integration points u based on the newly solved nodal pressures by:"*

---

## Author Comment (AC2)

We appreciate the anonymous referee for reviewing our manuscript. The comments encouraged us to advance the content and clarity of our study. Please find below a point-by-point response to the referees' comments (comments of the reviewer in black and our response in red)

On behalf of all authors,

Yours sincerely,

Maximilian Oskar Kottwitz

I have read and reviewed the manuscript "Equivalent continuum-based upscaling of flow in discrete fracture networks: The fracture-and-pipe model" by Kottwitz et al. The purpose of the study is to add the effects of "intersection flow localization" into equivalent continuum models. The authors argue that these localized effects are important to accurately capture the effective permeability and present several test cases. Overall, I found the presentation of the paper to be slightly confusing and am not left with a motivation to integrate such a model into my own continuum model as it seems like a solution to a problem that wouldn't exist in practical applications.

A few high-level comments:
This could be my own ignorance but I'm not clear on why you would use an ECM model over a DFN model with an impermeable matrix in the first place. Modern computing power allows us to simulate DFN models with thousands (and more) fractures. At a certain number of fractures, the ECM will actually be harder to accurately simulate because the resolution needed to preserve the topology of the DFN will quickly approach very low mesh sizes as the fracture network density increases. While some metrics might actually not be affected by high network density and artificial connections in the ECM, transport metrics like breakthrough of solute certainly would. Furthermore, there is no comparison between the ECM models presented here and DFN models in terms of effective permeability. Could that be used as an additional point of comparison rather than to the authors' own model? Previous ECM models (many cited within this paper) have been compared favorably (or unfavorably) to DFNs.

From our point of view, the fact that ECM's produce similar effective permeabilities as DFN's has sufficiently been demonstrated by the studies of Leung et al., 2012, Hadgu et al., 2017, and Sweeny et al., 2020 (all cited in this paper). We thus haven't seen the necessity to repeat this work in our studies. Our main intention rather was to promote existing ECM methods by incorporating the effects of IFL and local permeability anisotropy in a massively parallelized manner.

Due to the demonstrated comparability, we think that choosing an ECM or a DFN method for flow-based upscaling in fracture networks is mainly dependent on the preference/experience of the user, the available computational resources (e.g., single workstations or HPC clusters), and the natural application (deterministic DFN structure or stochastic DFN's). In our opinion, the big advantage of massive parallelization (up to 10'000s of cores) with efficient multigrid solvers make continuum methods highly suitable for situations, where very large numbers of simulations must be conducted (i.e., in Monte-Carlo-like sensitivity studies or potentially in adjoint based inversion approaches). For example, with our parallelized approach, ECMs with a resolution of 256 cubic voxels can routinely be solved on a decent workstation (here, 16 cores) in about a minute. This resolution was enough to adequately capture the structure of our test DFNs from section 6 (10000 fractures). If higher fracture densities are considered, we can routinely simulate ECMs with a resolution of 1024 cubic voxels on HPC systems (with 512

cores), which is enough to capture 3 orders of magnitude size scaling of fractures and very high fracture densities (e.g, with an average fracture spacing to system size ratio of 1e-4).

Yet, we agree that a direct comparison of effective permeabilities obtained by our and the DFN method would indeed be interesting, but not feasible given the brief time period (4 weeks) we received from Solid Earth to make corrections to our manuscript. However, if you know about existing DFN benchmark studies, we are willing to compare (1) the time spent on meshing of and simulating flow in a DFN structure as well as (2) the effective permeabilities resulting from DFN-software packages to our presented approach, favorably for DFNs with very high fracture densities (~ 100'000 fractures).

My biggest issue with this paper is the actual verification/validation of the IFL scheme. I encourage the authors to think about a way to clearly articulate: (1) when the IFL matters, (2) why it matters, and (3) and how much it matters (in a quantitative way). As it stands, there's some discussion points related to these, but they're not sufficient. In general, the authors have shown that a difference exists (e.g., in Figure 4), but not shown that this difference is indeed physically correct. I found section 5 to be very convoluted and not convincing with the self-comparison.

This point was also raised by the first reviewer, and we fully agree on the necessity to clarify the scale effects of IFL in a quantitative manner. In section 5, we have demonstrated that IFL increases the permeability of systems at the local ECM scale (system size of ~10times the fracture aperture, i.e., at fracture scales) if the intersection aligns with the applied pressure gradient. We did this by comparing the z-component of an analytically derived permeability tensor to the numerically computed permeability from high-resolution Stokes simulation (the benchmark). To explore the effects of IFL on larger scales (scales that cannot be resolved with current imaging techniques, i.e. larger than a few decimeters), we conducted effective permeability computations with the method described in the appendix on ECM's of several test-DFN's discretized with (1) the ECM method presented in section 4 (fracture-and-pipe method) and a fracture only ECM method (i.e., the method from section 4 without the intersection parametrization), similar as the ECM methods presented by Hadgu et al., 2017 or Sweeny et al., 2020. We added a new section in the revised manuscript that describes the details of this parameter study (section 6):

With this section, we hope to properly address the points indicated by both reviewers. We identified orthogonal fracture formations (so-called cross joints) as a natural candidate where IFL potentially matters most. With the conducted parameter study, we quantified the effects of IFL at these systems with various scales, apertures, fracture densities and matrix permeabilities. The general conclusion of this is that only for systems that have fracture porosities in the range of 30% (P33 is a non-dimensional quantity that combines all of the above-mentioned measures), IFL effects on the effective permeability become noticeable.

This lead to following changes in the abstract, starting from Line 13:

[revised manuscript text omitted]

The relationship between section 6 and the previous parts of the paper is not clear. My interpretation of section 6 is that it is showing the needed resolution of the ECM mesh size to have a solution converge. I think this is interesting, but not clear how this is related to the effects of the IFL. Seems like two different papers here.

In our opinion, checking the consistency of an employed method is inevitable in scientific works, especially if problems like resolution dependency are well-known to exits. To clarify this, we changed the beginning of section 6 (Lines 240 -244) to:

*"The resolution dependency of ECM methods to upscale the permeabilities of fracture networks is a crucial aspect that has to be considered to provide accurate upscaling results. Artificial connectivity is one of the main issues that arises, if the resolution of the ECM is insufficiently low as demonstrated by Jackson et al., 2000 and Svensson et al., 2001. Their results suggest that we can expect accurate upscaling results, only when the resolution of the ECM is sufficiently large to resolve the structure of the DFN (i.e., a maximum of two fracture segments and one intersection per cell)."*

We also changed the title of former section 6, now section 7 to:

*"Resolution dependency of ECM methods"*

Some more specific comments:
L42: With modern computing power, DFNs are commonly used at reservoir scale. I'd argue the difficulty is including matrix properties in DFNs/DFMs and meshing multidimensional DFMs.

We changed the sentence at Line 42 to:
*"Improved discretization techniques with individual fracture treatment (DFN method), inclusion of matrix properties in multi-dimensional meshes (discrete fracture and matrix - DFM - method) or multi-continuum methods come at the cost of high computational expenses."*

L74: Does a local increase in effective permeability really matter? Isn't the global effective permeability usually the quantity of interest? Which is being reported here?

You're right, the effective permeability of the system is the quantity of interest. We changed the sentence to:
*"As a consequence, the systems effective permeability should increase by a certain amount due to a local permeability increase within the intersection."*

I think Figure 5 is mentioned in text before Figures 2—4.

We changed the wording at Line 110 to:
*"… (as will be shown later by a comparison of errors to the result at largest resolution in plot b, figure 5)"*
If this still represents an issue, we can simply remove the line at this stage.

L155: Not all ECM models use regular grid spacing. Ref L575 used octree grids.

Indeed, Sweeney et al., 2020 use octree refinements for their grids. We changed Line 155 to:

*"The goal of the ECM method is to generate a 3D of computational cells that contains a symmetric, positive definite permeability tensor that is based on the fractures and their intersections. For simplicity, we prescribe a regular grid with constant x-y-z spacing dx instead of octree refined grids as utilized by Sweeney et al., 2020."*

L175-185: I'm not following how the calculation of $V_f$ is completed regarding this point counting. Why not just directly compute the intersection areas of the fractures with the cells? It would be more accurate. Also why is the z entry in the permeability tensor (13) 0 and not 1? Is the matrix permeability within an intersected cell taken into account?

Calculating the exact intersection areas of each fracture with a cell would of course be more accurate. However, it is very expensive from a computational perspective to calculate the intersection polygon of a disc with a 3D box, especially if cell size is very small compared to the fracture size. The point counting approach we use is much faster from a computational point, because we simply represent each fracture by a grid of equally spaced points and find the respective fracture points that belong to each computational cell. By multiplying the area of the grid spacing ($dp^2$) with the hydraulic aperture we get the volume fraction for the specific subregion of the fracture that is represented by the point. Multiplying this with the number of points per cell gives the total fracture volume per cell. Finer point spacing (dp) of course results in a better fit, which we demonstrate in figure 4, plot c.

The z-entry in the permeability tensor is zero, because we assume an anisotropic permeability tensor for a fracture that is aligned with the x-y plane. In that case there is no connectivity in the z-direction and hence no permeability. This approach results in the same permeability tensor as produced by the approach of Chen (1999). In eq. 14 we assume the same for an intersection that is aligned with the x-direction, hence no connectivity and permeability in the y- and z-direction.

Figure 4: I don't understand the point of this plot (c). What is the purpose of showing the norm of the tensor? It's showing a difference between fracture & pipe and just fracture, but what should we glean from this? What is the ground truth for comparison? How do we know which one is correct? Too much is being assumed of the reader here and elsewhere. Take us through the key points. There's interesting work here, but it's hidden in lots of verbosity.

The purpose of plot c in figure 4 is to demonstrate the quantitative difference in permeability magnitude of the resulting permeability tensors for a single cell ECM disretized with both methods, the fracture-and-pipe and the fracture-only approach. We've chosen to compare the norm of both tensors, as they also incorporate the difference of the off-diagonal components of the tensor.
Hence, the main point we want to make here is that we can actually have a large difference in permeability at the local fracture scale between both discretization methods, which is also evident by looking at the visualized tensors in plot b.
Admittedly, we haven't described the fracture-and-pipe and the fracture-only in the caption which is why this part can be confusing. We therefore expanded the caption of figure 4:

*"c) presents the norm of the permeability tensor $K_{ijk}$ as a function of the ratio between the ECM grid spacing dx and the initial point spacing dp for the discretization approach described in section 4 (the fracture-and-pipe model) and an approach, where we didn't take the IFL parametrization into account (i.e., leaving out eq. 14 in the discretization procedure, hence the name fracture-only)."*

The only assumption we stress in that section is the applicability of the point-counting method to approximate the cell volume fraction to rescale the permeability in the analytical ECM approach. By showing the convergence of the point-counting technique we argue that we properly addressed this assumption: With finer point spacing dP, we produce converged permeability values. I.e., if we use a cell size to point spacing ratio of 16, we are confident that the volume is sufficiently approximated by our point-counting approach to produce converged permeability results. We also corrected a typo in the x-label of figure 4, which should be written the other way around, i.e., dx/dp instead of dp/dx. We have changed that in the revised manuscript an in Line 211.

The presentation of section 5 is not very clear. What exactly is the benchmark solution being used for comparing the two different ECM approaches? How was this error metric (eq. 15) chosen – isn't this just a measure of convergence – why is it being compared between the two ECM approaches? The way I read this is that the method in section 2 is being used as the benchmark. So my two questions are then: (1) has the solver in section 2 itself been verified? And (2) Doesn't the method in section 2 include the "fracture & pipe" model (or is it DNS?)? Meaning that comparing the ECM without the intersection permeability will of course yield a worse fit.

We use the results of our direct flow approach (section 2, finite difference Stokes-flow with LaMEM) in resolved pore spaces of simple fracture crossings (e.g. figure 1) as the benchmark solution, i.e. the "true" value (k1024 in eq. 15). We are confident, that this represents the most accurate permeability result we can obtain for these simple fracture intersection models, as the software has been benchmarked for pore and fracture-scale cases in the studies of Eichheimer et al., 2019 and Eichheimer et al., 2020 (both cited in the paper), and also includes a large number of Stokes benchmarks as part of the testing suite. In both papers, we could reproduce the results of several analytical solutions, analog permeability measurements on the Fontainebleau sandstone and sintered glass beads with our numerical simulations. In all cases, the numerical resolution of the direct flow simulations should be sufficiently high (usually 1024 cubic voxels, which are computationally quite expensive as they involve HPC simulations on 8000 cores). We conducted the same type of HPC simulations here and showed the convergence of the result with increasing resolutions (plot b in figure 5). With an average error of the result of 0.6% for the last step of resolution increase (512 to 1024 cubic voxels), we assume that the result is sufficiently converged and represents an accurate solution of the systems permeability.

Thus, we use this solution as the true benchmark value that is compared to the analytically derived, single cell ECM permeabilities (kx in eq. 15) that result from the fracture-and-pipe and the fracture-only discretization approach. By this, we can test the accuracy of the ECM method, which, to our knowledge, has not been benchmarked elsewhere. The chosen error metric (L2 error norm) is indeed often (actually as well in figure 5) used to demonstrate numerical convergence. However, it is also a good indicator for the accuracy of a prediction, as it quantifies the error or absolute difference of one quantity with respect to another.

The one thing I noticed throughout this paper is the fracture apertures are quite large (mm to cm). Most practical applications the authors listed in the beginning of the paper can include fractures with much smaller apertures. How will this affect the influence of the IFL? This might be interesting to investigate here.

We incorporated quite small apertures (sub-millimeter) in the parameter study in the new section 6 by utilizing a prefactor of 1e-4 in the sub-linear aperture-length correlation model. As already mentioned above, the influence of IFL for small apertures was not as significant as for fracture systems with larger apertures.

Just a note that it's not "DFN's" and "ECM's", but "DFNs" and "ECMs"

We changed the spelling accordingly throughout the paper.

L262: I think this is the first mention of matrix permeability in the paper. How is the matrix permeability included in the fracture intersection cells if their dimensions are larger than the fracture apertures? Also, how do the effects change with different matrix permeabilities?

Matrix permeabilities are prescribed as isotropic (constant $k_{xx}$, $k_{yy}$ and $k_{zz}$ values) in the computational grid prior to adding fracture and intersection permeabilities to respective cells. We assume that the prescribed value is already properly rescaled to the cell volume. We incorporated different matrix permeabilities (1e-17, 1e-15, and 1e-13) in the parameter study of the new section 6. There was no noticeable dependence of the IFL effect on matrix permeability.

What are the readers supposed to learn from Figure 9? That changing the cell size changes the permeability? That's obvious based on the upscaling equations. Please explain further. I think it's showing the effects of artificial connections in the ECM, but again, this doesn't mean it's physically correct. It just means it's converging. Please expand on the discussion here.

With this figure and section, we wanted to demonstrate the resolution dependency of ECM methods. We'd argue that if one wants to upscale a structure properly based on an ECM, the permeability should not change as a function of cell size but rather be consistent throughout scales.

The changes for large cell sizes are indeed caused by artificial connectivity. However, they vanish at sufficiently high resolutions. So if the resolution is sufficiently high and artificial connectivity is avoided, we have demonstrated that we actually produce constant/converged permeability tensors with our ECM method. I.e., the permeability does not change anymore with cell-size changes, and which we argue to be the most accurate ECM representation of a DFN.

Determining whether the resulting permeability tensor is physically correct is indeed a challenging issue. We guess that this can only be properly addressed in an extensive benchmark study with a comparison to natural field measurements, which we are actually working on at the moment. However, due to the seminal results of Leung et al., 2012, Hadgu et al., 2017 and Sweeny et al., 2020, we at least assume that the result is at least similar to the results of the DFN method.

In the same light, what is the point of section 6? The topic sentences in the beginning of the section promise some presentation of the accuracy of the method for large DFNs. I see no quantitative measures of error in this section. Just numerical convergence.

Please see previous answers. Using the term "accuracy" was indeed misleading and could cause confusion. We changed the title and the first sentences of the topic to highlight that we address the resolution dependency of ECM methods.